# Is attention required for ICL? Exploring the Relationship Between Model Architecture and In-Context Learning Ability

**Ivan Lee, Nan Jiang, Taylor Berg-Kirkpatrick**
University of California, San Diego
{iylee,n3jiang,tberg}@ucsd.edu

## Abstract

What is the relationship between model architecture and the ability to perform in-context learning? In this empirical study, we take the first steps toward answering this question. We evaluate thirteen model architectures capable of causal language modeling across a suite of synthetic in-context learning tasks. These selected architectures represent a broad range of paradigms, including recurrent and convolution-based neural networks, transformers, state space model inspired, and other emerging attention alternatives. We discover that all the considered architectures can perform in-context learning under a wider range of conditions than previously documented. Additionally, we observe stark differences in statistical efficiency and consistency by varying the number of in-context examples and task difficulty. We also measure each architecture's predisposition towards in-context learning when presented with the option to memorize rather than leverage in-context examples. Finally, and somewhat surprisingly, we find that several attention alternatives are sometimes competitive with or better in-context learners than transformers. However, no single architecture demonstrates consistency across all tasks, with performance either plateauing or declining when confronted with a significantly larger number of in-context examples than those encountered during gradient-based training.

## 1 Introduction

*In-context learning* (ICL) refers to the ability to learn new tasks at inference time, using only input-output pair exemplars as guidance. Radford et al. (2019) demonstrate early signs of this ability in GPT-2, a causal transformer (Vaswani et al., 2017). ICL was further popularized by GPT-3 (Brown et al., 2020), a large language model with the same architectural foundation but augmented with greater capacity and trained on large-scale data. By simply adjusting a natural language prompt, it was shown that GPT-3 could adapt to new tasks, such as translation and question answering, without updating any of its parameters. These findings spurred significant interest in the research community to investigate this curious behavior (Zhao et al., 2021; Min et al., 2022; Liu et al., 2022).

Yet, a prevailing uncertainty remains: are large language models genuinely learning from their prompts or simply being conditioned to surface relevant aspects of their training data? To address this, a new line of research emerged that examines ICL in controlled, synthetic environments where task resolution fundamentally depends on prompt utilization (Xie et al., 2021; von Oswald et al., 2022; Garg et al., 2023; Akyürek et al., 2023). However, most of these studies anchor their investigations on the assumption that models utilize an internal attention mechanism (as is the case for transformers). Whether attention mechanisms are necessary for in-context learning to emerge remains an open question.

Notable exceptions to this assumption include Xie et al. (2021) and Chan et al. (2022) who consider recurrent neural networks alongside transformers. The former finds RNNs and LSTMs fail to learn image classification in the ICL setting. In contrast, the latter demonstrate that LSTMs possess ICL abilities in a synthetic language modeling task, where hidden Markov models generate the data.

Table 1: Examples of our synthetic in-context learning tasks.

| Task | Prompt | Target | |
|---|---|---|---|
| Associative Recall | a, 1, b, 3, c, 2, b | 3 | |
| Linear Regression | $\mathbf{x}_1, y_1, \mathbf{x}_2, y_2, \mathbf{x}_3, y_3, \mathbf{x}_4$ | $y_4$ | $\exists \mathbf{w}$ such that $\forall i, y_i = \mathbf{x}_i \cdot \mathbf{w}$ |
| Multiclass Classification | $\mathbf{x}_1, b, \mathbf{x}_2, a, \mathbf{x}_3, a, \mathbf{x}_4$ | b | $x_1, x_4 \sim \mathcal{N}(y_b, I_d)$ 
 $x_2, x_3 \sim \mathcal{N}(y_a, I_d)$ |
| Image Classification | ✿4 🧍9 🧍9 ✿4 ✿4 🧍9 ✿ | 4 | bursty training prompt |
| | ⊙5 △8 🧍9 🐎6 ⊢3 ✿4 🦤 | 2 | non-bursty training prompt |
| | 🦤1 🐚0 🐚0 🦤1 🦤1 🐚0 🐚 | 0 | evaluation prompt |
| Language Modeling | *Colorless green ideas sleep* | *furiously* | |

However, whether both findings are specific to their task or indicative of more general behavior remains uncertain.

The community's focus on attention is understandable given the success of transformers. However, the architecture comes with a number of limitations, such as quadratic time and memory complexity. These limitations spurred research into alternative architectures such as efficient self-attention models (Tay et al., 2022a) and state space models (Gu et al., 2021). If these alternatives are to replace transformers as the dominant model architecture, it is natural to wonder if they are capable of ICL. Moreover, some are designed to handle prompts of arbitrary length, potentially introducing a novel ICL form, constrained only by dataset size rather than inherent architectural limitations. Furthermore, classic architectures such as recurrent neural networks and convolutional neural networks were once the backbone of machine learning research before the introduction of transformers and ICL as a concept. Do these classic architectures inherently lack ICL capabilities, or were they simply constrained by the compute and data available during their heyday.

In this study, we set out to address the aforementioned questions. Specifically, we aim to answer the following research questions: *Which architectures are capable of ICL, and which exhibit superior ICL performance?* Our primary focus lies on the former question. While the latter is more challenging to assess, our experiments provide insights into which families of architectures tend to perform well, even if they do not offer definitive answers. To advance our objectives, we evaluate a diverse range of model architectures that span several design paradigms. This includes both the classical methods previously mentioned and modern approaches such as the transformer and those inspired by state space models. Our assessment covers the ICL capabilities of each architecture over a wide array of synthetic tasks, spanning different modalities and including both classification and regression, as depicted in Table 1.

Our specific contributions are as follows:

- LARGE-SCALE EMPIRICAL STUDY: We conduct the first large-scale empirical study comparing ICL performance across diverse model architectures, shedding light on their relative strengths and weaknesses. Code is available at `https://github.com/ivnle/synth-icl`.

- UNIVERSALITY OF ICL: We discover that all the considered architectures can perform in-context learning under a wider range of conditions than previously documented, lending support to the position that ICL is not exclusive to attention-based models.

- EMPIRICAL SUCCESS OF ATTENTION ALTERNATIVES: Our findings demonstrate that some attention alternatives not only compete with but, in certain cases, surpass transformers at in-context learning. This suggests that efficiency gains in these architectures do not necessarily come at the expense of performance.

## 2 SYNTHETIC IN-CONTEXT LEARNING TASKS

Studying in-context learning in large language models presents inherent challenges. One fundamental question is whether these models are truly learning new predictors during the forward-pass, or whether in-context examples simply focus the model on specific aspects of the knowledge already acquired during gradient-based pretraining. While from a Bayesian perspective this dichotomy represents endpoints of a spectrum (Xie et al., 2021), it nonetheless clouds interpretation of ICL experimental results. To address this concern, a new line of research has emerged that examines ICL in controlled, synthetic environments where task resolution depends fundamentally on prompt utilization (von Oswald et al., 2022; Garg et al., 2023; Akyürek et al., 2023). In these settings, models must rely on their prompts to solve tasks, eliminating the possibility of memorization: Models are trained from scratch to take a labeled dataset as input and then predict the result of learning from this data directly in the forward-pass of the resulting model. Thus, each train and test example is a unique learning problem but of a consistent type (e.g. linear regression).

In addition to offering a clearer perspective on in-context learning, synthetic tasks have low computational requirements. These decreased barriers allow for more equitable comparisons across model architectures. Utilizing publicly available pretrained models may introduce confounding variables, stemming from disparities in model capacity, training durations, and data quality. By training models from scratch on synthetic tasks, we are given greater control over these factors. Furthermore, a suite of such tasks is a valuable tool for the research community, enabling rapid benchmarking of emerging architectures without the intensive computational overhead typically associated with large language models.

For these reasons, we curate a suite of synthetic in-context learning tasks and summarize them in Table 1. The majority of our tasks take the form

$$\underbrace{x_1, f(x_1), x_2, f(x_2), ..., \overbrace{x_n}^{\text{query}}}_{\text{prompt } P}, \underbrace{f(x_n)}_{\text{completion}}$$

where the goal is to learn function $f$ by observing a *prompt*, a sequence of input-output pairs $(x_i, f(x_i))$, which ends with a *query*. The model's objective is to produce an appropriate *completion* based on the given prompt. We train model $M_\theta$ parameterized by $\theta$ to minimize the expected loss over all prompts

$$\min_\theta \mathbb{E}\left[\ell\left(M_\theta(P), f(x_n)\right)\right], \tag{1}$$

where $\ell(\cdot, \cdot)$ is the appropriate loss function for a given task.

**Associative recall** (Ba et al., 2016; Fu et al., 2023) is the task of learning key-value mappings from a prompt and can be viewed as the simplest form of in-context learning. Let $V$ be a discrete vocabulary of size $k$. We consider the class of functions

$$F = \{f | f : V \xrightarrow{\text{B}} V\}$$

where $f$ is a bijective mapping. These mappings are created by randomly pairing elements of $V$ without replacement, ensuring each element maps to a unique counterpart. We uniformly sample $f$ from $F$ and $x_1, ..., x_n$ from $V$ to construct the prompt as $P = (x_1, f(x_1), x_2, f(x_2), ...x_n)$. Elements of $P$ are mapped to vectors with a simple lookup table, as is standard in language modeling.

**Linear regression** (Garg et al., 2023) is the task of learning a linear function from a prompt. We consider the class of functions

$$F = \{f | f(x) = \mathbf{w}^\top x, \mathbf{w} \in \mathbb{R}^d\}$$

We sample $x_1, \ldots, x_n$ and $w$ from the isotropic Gaussian distribution $\mathcal{N}(0, I_d)$. We then compute each $y_i = \mathbf{w}^\top x_i$ and construct the prompt as $P = (x_1, y_1, x_2, y_2, \ldots, x_n)$. Since $y_i$ is a scalar, we represent it as a $d$-dimensional vector, with its first index set to $y_i$ and remaining indices set to zero.

**Multiclass Classification** is a clustering task in which the items to be clustered are sampled from $k$ distinct Gaussians. For this task, we use the procedure

$$\mu_i \sim U(-1, 1)^d, \text{ for } i = 1, \ldots, k$$

$$y_j \sim U(\{1, \ldots, k\}), \text{ for } j = 1, \ldots, n$$

$$x_j \sim \mathcal{N}(\mu_{y_j}, I_d), \text{ for } j = 1, \ldots, n$$

to construct the prompt as $P = (x_1, y_1, x_2, y_2, \ldots, x_n)$. Since $y_j \in \{1, \ldots, k\}$, we map each cluster label to a $d$-dimensional vector with a simple lookup table. We set $d$ to 16 in all experiments.

To facilitate a clearer understanding, we defer detailed discussions of **Image Classification** and **Language Modeling** to Sections 5 and 6, respectively.

## 3 MODEL ARCHITECTURES

**Recurrent**   We consider three common variations of recurrent neural networks: Elman (Rumelhart et al., 1986, **RNN**), long short-term memory (Hochreiter & Schmidhuber, 1997, **LSTM**), and gated recurrent unit (Cho et al., 2014, **GRU**). Recurrent neural networks are characterized by their length-invariant inference cost and theoretically infinite context size, though empirical findings suggest an upper limit on this context size (Khandelwal et al., 2018). Furthermore, since the introduction of transformers, this class of architecture has seen diminished focus within the community, particularly in the ICL setting. We believe revisiting approaches that have fallen out of favor helps counterbalance the community's potential over-reliance on a select few contemporary methodologies.

**Convolutional**   Representing the class of convolutional neural networks (CNN), we focus on the architectures proposed by Wu et al. (2019): lightweight convolutions (**LIGHTCONV**) and dynamic convolutions (**DYNAMICCONV**). These architectures, derived as special cases of depthwise convolutions (SIfre & Mallat, 2014), have demonstrated competitive performance with transformers in specific contexts (Tay et al., 2022b). LIGHTCONV is simply a depthwise CNN with weights normalized across the temporal dimension via a softmax. This design means that, unlike in self-attention, its context window is fixed and the importance placed on context elements does not change across time. To remedy this shortcoming, DYNAMICCONV predicts a different convolution kernel at every time-step. However, the kernel is a function of the current time-step only as opposed to the entire context as in self-attention. Similar to the recurrent class, CNNs exhibit length-invariant inference costs. However, they trade infinite context size for training parallelism.

**Structured State Space Sequence Models (SSMs)**   We also examine a category of recently proposed architectures inspired by state space models (Kalman, 1960). These architectures attempt to merge the efficient inference capabilities of RNNs with the parallel training attributes of transformers and CNNs. **S4** (Gu et al., 2021) set a new state-of-the-art on long-range sequence modeling, but falls short in language modeling compared to transformers. Subsequently, **H3** (Fu et al., 2023), **HYENA** (Poli et al., 2023), and **Mamba** (Gu & Dao, 2023) were proposed, each progressively improving upon this language modeling gap. We also include architectures inspired by linear attention (Katharopoulos et al., 2020; Zhai et al., 2021). Specifically, we examine **RETNET** (Sun et al., 2023) and **RWKV** (Peng et al., 2023). While not necessarily inspired by state space models, these architectures also strive for efficient inference, parallelizable training, and can be viewed as variants of SSMs.

**Transformers**   Finally, we consider two popular autoregressive transformer designs: **GPT2** (Radford et al., 2019) and **LLAMA2** (Touvron et al., 2023). Their primary differences lie in choice of positional embeddings and activation functions. GPT2 utilizes learned absolute positional embeddings and ReLU activation while LLAMA2 incorporates rotary positional embedding (Su et al., 2022) and SWIGLU activation (Shazeer, 2020). Rotary embeddings endow transformers with both absolute and relative positional information through rotations in complex space. We also perform an ablation study across positional embeddings (or lack thereof) and show our results in Appendix E.

Note that we train all models from scratch, adopting only the architectural design choices made by the named models' authors. In the following sections, we delve into our experimental methods and findings. Section 4 presents our results for linear regression, associative recall, and multiclass classification. We discuss image classification outcomes in Section 5, and conclude with our language modeling results in Section 6.

# 4 LEARNING TO LEARN (IN-CONTEXT)

In our initial experiments, we evaluate the capacity of various architectures to in-context learn associative recall, multiclass classification, and linear regression. Results are shown in Figure 1 and experimental details are shown in Appendix A.1. Besides confirming the existence of ICL ability, we are particularly interested in measuring *statistical efficiency*—which models make better use of a fixed amount of data (in-context examples)—and in determining if our trained models demonstrate *consistency*, i.e., whether their performance converges in probability to some ceiling.

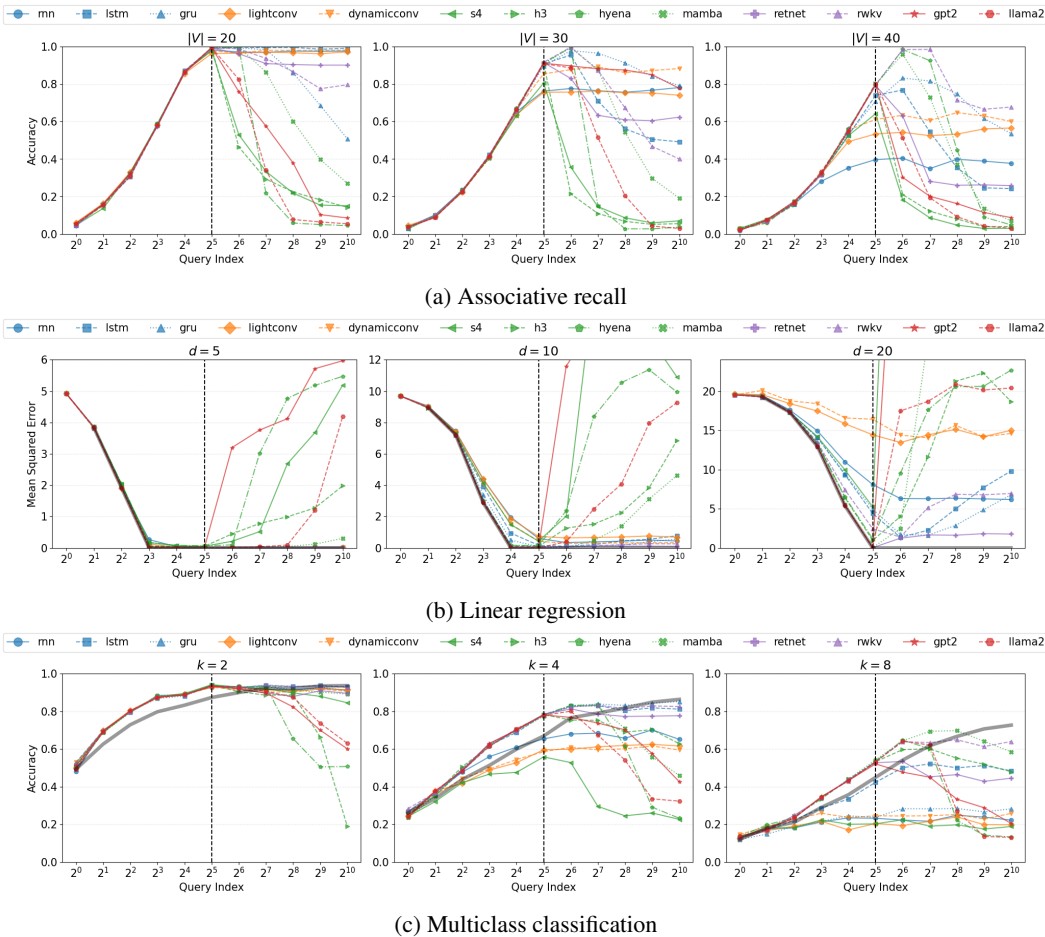

(a) Associative recall

(b) Linear regression

(c) Multiclass classification

Figure 1: *Evaluating various architectures on associative recall, linear regression, and multiclass classification.* We plot test accuracy and mean squared error as a function of the number of in-context examples. A query index of $2^5 = 32$ implies 31 in-context examples, which is also the highest number of in-context examples seen during training (vertical dotted line). Task difficulty increases from left to right. Each line represents the single run that achieved the best validation accuracy or mean squared error at query index $2^5$. See Tables 9, 7, 11 for a tabular view of the same data. See Figure 5 for average performance across training runs. See Appendix B.1 for linear regression experiments with Gaussian noise where we observe trends are largely unchanged relative to the non-noisy setting. Classical baselines (black) are shown for linear regression (ridge regression) and multiclass classification (logistic regression).

Why is consistency of interest? First, a proficient learner, irrespective of the ICL setting, is expected to improve its performance given more i.i.d. training data. Consequently, a rise in in-context examples should lead to regular performance improvements. However, it is unclear if this is true in the in-context setting, a query we offer clarity on shortly. Second, the emergence of length-invariant inference architectures, rivaling transformers in task performance, paves the way for ICL with a substantially larger number of in-context examples than what is typically used today. One can imagine

a new paradigm to replace finetuning: adapting pretrained language models to new tasks by utilizing a precomputed (previous) hidden state without parameter updates.

**All architectures can in-context learn.** We first turn our attention to the left most plots in Figure 1, and specifically the region left of the dashed vertical line. Clearly, all architectures successfully in-context learn the three tasks. This provides an existence proof that ICL is not a unique property of transformers. Differences among the architectures becomes more evident as we increase difficulty and take into account their ability to extrapolate to large data sizes than seen in training (right of the dotted vertical line).

**Which architectures are consistent?** Initially, all architectures appear consistent when considering only prompt lengths encountered during training. However, this perception changes when we introduce prompt lengths well beyond those seen during training. Specifically, the performance degradation is most pronounced in the four state space model inspired architectures and the two transformers. Note that this behavior is expected for GPT2 which uses learned positional embeddings, but not for LLAMA2 which uses rotary embeddings. Interestingly, other architectures with recurrent formulations (such as the RNNs, RETNET, and RWKV) do not exhibit such drastic declines. This also holds true for the CNNs, which are inherently limited to finite context lengths. This behavior in CNNs makes intuitive sense, as long range information that may "confuse" this architecture class are discarded over time. It is possible that, similar to RNNs (Khandelwal et al., 2018), RETNET and RWKV exhibit stronger preference to nearby context relative to the state space model inspired architectures (originally motivated by long sequence modeling) and transformers (which have random access to their entire context). This preference may explain why these architectures are more robust to unseen prompt lengths.

**Variations in statistical efficiency.** The following summary assumes the most difficult setting for all tasks. For associative recall, the top performers were the transformers, H3, HYENA, MAMBA, RETNET, and RWKV when given 31 in-context examples (the longest prompt length seen during training). When extrapolating to longer prompt lengths, HYENA, MAMBA, and RWKV achieved near perfect accuracy, but performance degraded as the number of in-context examples grew. Our ablation over positional embeddings in Table 15 reveal that transformers without positional embeddings and transformers with sinusoidal embeddings are the best at associative recall regardless of prompt length. For linear regression, the transformers, MAMBA, and RETNET achieve near perfect MSE when given 31 in-context examples. Interestingly, these four architectures match the performance of ridge regression. Beyond 31 examples, however, performance quickly deteriorates, with RETNET showing the most robustness to this deterioration. Surprisingly, GRU and LSTM demonstrated competitive performance when extrapolating to unseen prompt lengths. We saw improved extrapolation ability in transformers without positional embeddings (Table 16), but its performance still degraded as the number of examples increased. For multiclass classification, the transformers, all the state space model inspired architectures (except for S4), RETNET and RWKV achieved the best accuracy, surpassing logistic regression. In particular, MAMBA scored the highest accuracy when given 255 in-context examples. We also note that LSTM was competitive with the other architectures but did not achieve a top score.

**Hyperparameter sensitivity.** We now consider *average* performance for each architecture (Figure 5). Earlier, we found that some RNNs, despite not achieving the best scores, were competitive with modern architectures. However, these performances were difficult to replicate and were isolated to a few lucky combinations of hyperparameters. For associative recall, the transformers, HYENA, MAMBA, and RETNET were consistently strong performers. In particular, MAMBA achieved an average accuracy of 0.96 when given 63 examples. For linear regression, LLAMA2 was the clear leader for prompt lengths seen during training, followed by RETNET. For multiclass classification, LLAMA2, MAMBA, and RWKV were the top performers, followed by H3 and HYENA. Both RWKV and MAMBA improved in performance as prompt lengths increased beyond those seen during training. Interestingly, multiclass classification was the sole task where GPT2 did not perform well on average.

## 5 THE INFLUENCE OF TRAINING DATA DISTRIBUTIONAL PROPERTIES

We now study how the distributional properties of training data can influence ICL. We follow the image classification experiments of Chan et al. (2022) who show ICL emerges when training data

exhibits particular properties such as burstiness and having large numbers of rarely occurring classes. To manage the number of experiments in this study, we focus exclusively on burstiness, a feature of natural data not found in typical supervised datasets. For example, natural language is temporally 'bursty''. That is, a given entity (e.g., word, person) may appear in clusters rather than uniformly across time (Altmann et al., 2009).

We train models on a mixture of *bursty* and *non-bursty* prompts. See Table 1 and Figure 7 for examples. In bursty prompts, the query class appears 3 times. To prevent the model from simply outputting the most common class in the prompt, a second class also appears 3 times. Bursty prompts can be solved by either leveraging query-label pairs across *different* training prompts (i.e. memorization) or referring to the in-context examples within prompts (i.e., ICL). For non-bursty prompts, the image-label pairs are drawn randomly and uniformly. This implies there is no incentive for a model to utilize the in-context examples. Note that models now have two options to learn how to classify images: memorization or ICL. This stands in contrast to our experiments in Section 4 where ICL was the only option to solve a task. We want to understand if certain architectures are predisposed towards adopting one of these modes.

We evaluate models with standard few-shot sequences containing images from two holdout classes and randomly assign one class to label 0 and the other to label 1. To solve this evaluation task, the model must utilize ICL. Images are sourced from Omniglot (Lake et al., 2019), a dataset of handwritten characters with 1623 classes. We follow Chan et al. (2022) and embed images using a randomly initialized ResNet (He et al., 2015) that trains alongside the evaluated model. Their corresponding labels are mapped to vectors with a simple lookup table. We perform the same sweep outlined in Section 4 resulting in 1512 training runs. We show our results in Figure 2 with supplementary results in Appendix C. We note that all training runs achieved near perfect training accuracy, confirming that models have indeed learned at least one of the two methods of image classification.

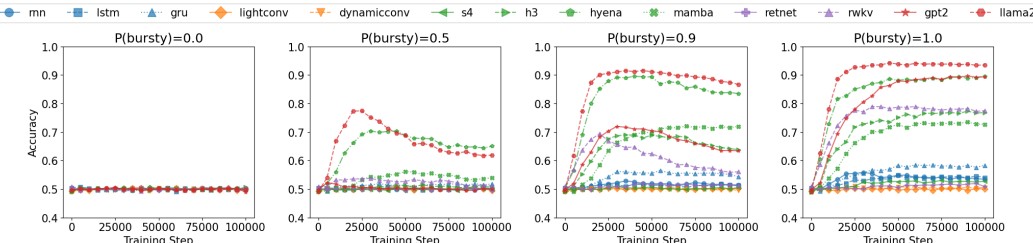

Figure 2: *Measuring the effects training data distributional properties on in-context learning.* We plot average (over training runs) test accuracy as a function of training steps. P(bursty) indicates the proportion of training prompts that were bursty (with the remainder non-bursty). See Table 14 for a tabular view of the same data. See Figure 8 for training runs that achieved max validation accuracy.

**Can ICL emerge given purely non-bursty examples?** As shown in the first column of Figure 2, no architectures demonstrate ICL ability when all prompts are non-bursty. This is not surprising given that i.i.d in-context examples rarely provide useful information for classifying the query image.

**Are some architectures predisposed towards ICL?** After increasing P(bursty) to 0.5, we find that LLAMA2 and HYENA demonstrate a strong preference towards ICL. It is surprising that GPT2 did not share this predisposition as it is similar in design to LLAMA2. We hypothesize that the rotary positional embeddings employed by LLAMA2 provide a stronger inductive bias towards ICL than the absolute learned positional embeddings used by GPT2. Further increasing P(bursty) to 0.9 reveals that ICL ability emerges consistently in GPT2, MAMBA, H3, and RWKV.

**Are some architectures predisposed towards memorization?** Setting P(bursty) to 1 reveals that a subset of architectures strongly prefer memorization over ICL. In particular, RETNET, S4, the two CNNs and all three RNNs strongly favor memorization. This is not to say that these architectures are incapable of solving this task which we address shortly. We were particularly surprised at the resistance of RETNET to develop ICL ability given that it was one of the top performers in Section 4. ICL emerged in only 2 of 108 training runs for RETNET, and notably, this development occurred after 30K training steps, a window similar to that of the three RNNs. In contrast, the other high-performing architectures from Section 4 developed ICL capabilities in fewer than 10K steps.

**Does ICL emerge in all architectures?**   While average accuracy across training runs is depicted in Figure 2, we also present the training runs that achieved the best validation accuracy in Figure 8. In these analyses, we observe that ICL emerges in all evaluated architectures, except for LIGHTCONV. We hypothesize that the absence of a time-step dependent kernel, a feature present in DYNAMIC-CONV, might be responsible for this outcome. Interestingly, ICL emerges in all three RNNs when P(bursty) is set to 0.9 and 1.0, a finding that contradicts those reported by Chan et al. (2022). More-over, GRU exhibits the ability to perform ICL even with P(bursty) set as low as 0.5. Given that the RNNs fail at this task *on average*, we credit this finding to luck with our hyperparameter sweep.

## 6   TOWARDS IN-CONTEXT LEARNING IN THE REAL WORLD

Up until now, our experiments have fallen under the few-shot learning concept of ICL where models are prompted with several in-context examples in a next-token-prediction format. We now consider an alternative perspective on ICL, represented in Kaplan et al. (2020) and Olsson et al. (2022). This approach focuses on observing loss at different token indices to measure improvements in language modeling performance as context length grows. Indeed, this is simply what language models are designed to do. However, as the their ability to predict later tokens based on earlier ones improves, they can be utilized in increasingly interesting ways, such as instruction following.

We report both *in-context learning score* and validation loss in Figure 3. Olsson et al. (2022) define in-context learning score as "the loss of the 500th token in the context minus the average loss of the 50th token in the context, averaged over dataset examples." One can view ICL score as a simple heuristic to measure the statistical efficiency of a given model. Note that this task is distinct from the large language model setting of in-context learning, where models are trained on language modeling and undergo evaluation with few-shot prompts. We assess models on the same task they were trained on: next-token prediction. See Appendix A.2 for experiment details.

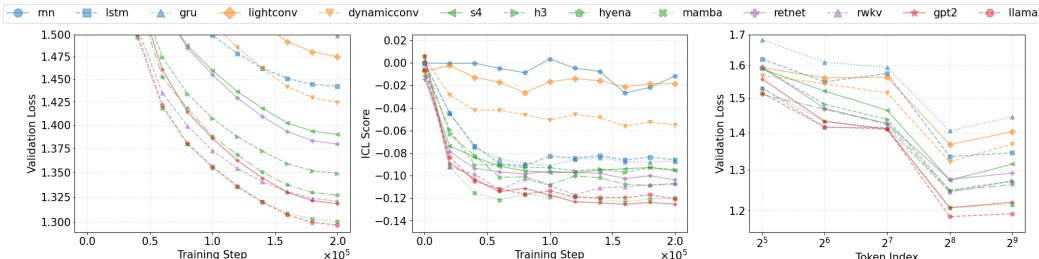

Figure 3: *Evaluating architectures on language modeling.* **Left:** Validation loss during training. **Middle:** ICL score as training progresses. **Right:** Validation loss as a function of context length.

**Most architectures exhibit an abrupt improvement in ICL score.**   This same phenomenon was noted by Olsson et al. (2022) in transformers. They discover that induction heads, which they hypothesize as the key mechanism behind ICL, form during the same window where ICL score abruptly improves. Since most architectures considered do not incorporate the concept of an attention head, an intriguing question emerges: What mechanism, analogous to induction heads in transformers, exists in these alternative architectures that facilitate a similar role in ICL?

**Does ICL score correlate with our previous experiments?**   In Section 4, our top performers included the two transformers, RWKV, RETNET, H3, HYENA, and MAMBA. Section 5 shares this list (except for RETNET). Consistently, these architectures also achieved the highest ICL scores, led by the transformers and MAMBA. We noted that DYNAMICCONV and LSTM, despite sharing similar validation loss, exhibited a significant gap in ICL score. We find that, when considering their best training runs, LSTM consistently outperformed DYNAMICCONV in all prior tasks and demonstrated superior extrapolation abilities. We observe the same relationship between GRU and LIGHTCONV. While ICL score does appear to correlate with performance in the previous sections, it should not be considered in isolation. For example, S4 and H3 share almost identical ICL scores. However, S4 did not perform as well in our prior tasks as H3 and achieved a lower validation loss on language modeling. Lastly, it is worth mentioning that RNN, despite its poor ICL score, outperformed the two CNNs in image classification when looking at their best training runs (see

Table 13). This suggests that RNN might be more effective at ICL than the CNNs in scenarios with shorter prompt lengths, as our image classification experiments used prompt lengths of 17 versus 512 in language modeling. We also observe that ICL ability in Section 5 appears to emerge during the same window where ICL score dramatically improves, lending credibility to Olsson et al. (2022)'s use of the metric.

## 6.1 A SIMPLE FEW-SHOT NATURAL LANGUAGE TASK

An interesting property of the dataset we use for language model training (Appendix A.2) is that we can produce relatively small models that still result in fluent language generation. To take advantage of this property, we evaluate architectures on a final ICL task that more resembles those used with large language models: in-context examples are composed using only natural language. Specifically, we compose 200 sentence pairs of the following form: *"Lilly scrapped her knee. Lily is sad."* Given a target number of in-context examples, for each of the 200 pairs, we randomly sample from the remaining 199 pairs without replacement to assemble 200 prompts. We ensure the two classes (happy and sad) are balanced. For example: *"Lilly scrapped her knee. Lily is **sad**. Lilly played with her friends. Lilly is **happy**. Lilly ate ice cream. Lilly is _____"*. This procedure is repeated 10 times yielding 2000 prompts for each target number of in-context examples.

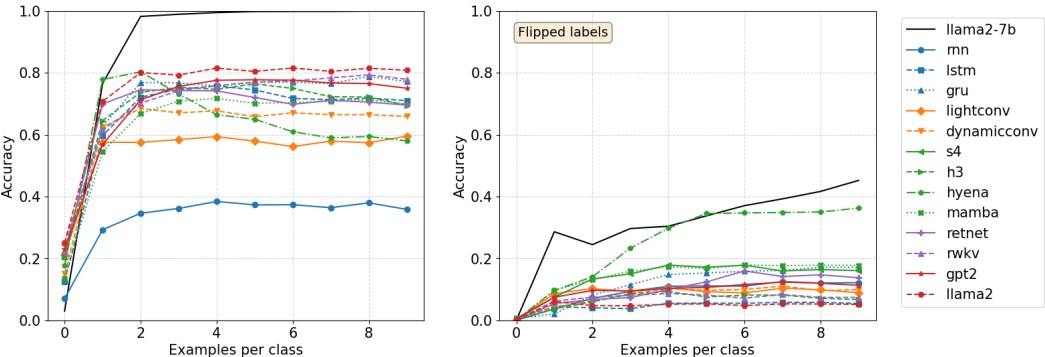

Figure 4: *Evaluating various architectures on a simple natural language ICL task.* We report accuracy as a function of the number of in-context examples. We use the open sourced weights for Llama2-7B and do not fine-tune. All other models are trained from scratch and are approximately 33M parameters (excluding embedding layers). **Right:** Flipped label setting, i.e., "happy" is replaced with "sad" and vice versa. See Figure 9 for normalized accuracy.

We also repeat the experiment but flip the classes, i.e., all instances of "sad" are replaced with "happy" and vice versa, testing if the model can override semantic priors (Wei et al., 2023). We show our results in Figure 4. Note that we include Llama2-7B as a reference point. We use the open sourced weights for this model as is and do not further train it on TinyStories.

**Accuracy improves with more examples, but quickly plateaus in the unflipped setting.** This pattern held true for all architectures, with the exception of HYENA which showed an initial peak in accuracy, followed by a decline. This decay was also noted in Section 4, when HYENA encountered prompt lengths unseen during training. However, the prompt lengths in the current context fall well within the sequence lengths encountered during their language model training. Given how quickly accuracy plateaus for all architectures, we believe that any gains are due to reallocating probability mass from non-target tokens to both target tokens, rather than truly learning in-context.

**Most architectures fail in the flipped setting.** A notable exception was HYENA, which demonstrated steady improvement up to 5 examples per class before plateauing. This suggests that HYENA, among the architectures we considered, might possess a stronger capability to override its semantic priors. However, we are unable to reconcile this with the observed performance decay in the unflipped setting.

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

## A EXPERIMENTAL DETAILS

### A.1 EXPERIMENTAL DETAILS FOR LINEAR REGRESSION, MULTICLASS CLASSIFICATION, AND ASSOCIATIVE RECALL

We train each model with prompts containing 32 in-context examples. Training loss is computed for each of the examples and averaged, i.e., models are effectively trained on prompts of varying lengths. We evaluate the trained models on prompts comprising 1024 in-context examples, assessing their ability to extrapolate to unseen prompt lengths. We train each architecture for 100,000 iterations with a batch size of 128. Embedding size is fixed to 64 but we sweep over 3 learning rates, 3 layer depths, 3 seeds, 3 difficulties and 3 tasks, for a total of 243 training runs per architecture (Table 2). Some architectures contain far less parameters per layer than others. For example the largest model trained was RETNET with 530K parameters while the largest GRU was only 200K parameters. To account for this discrepancy, we conduct 81 extra training runs for each of the smaller architectures by adjusting their embedding size and layer depth such that their parameter count is approximately 500K (Table 3).

### A.2 Experimental details for language modeling

We trained each architecture on 5.12 billion tokens of TinyStories (Eldan & Li, 2023), a synthetic dataset of short stories which contain only words that 3 to 4-year-olds typically understand. The stories are generated by GPT-3.5 and GPT-4 and summary statistics are presented in Table 6. All models were approximately 33 million parameters (excluding embedding layers). Unless otherwise specified in Table 5, we set embedding size to 512 and layers to 8. Additional settings and hyperparameters are shown in Table 4.

## B Supplementary data for Section 4: associative recall, linear regression, multiclass classification

We show line plots of average performance on associative recall, linear regression, and multiclass classification across all training runs in Figure 5. Tabular views for linear regression are shown in Tables 9, 10, associative recall in Tables 7, 8, and multiclass classification in Tables 11, 12.

### B.1 Noisy linear regression

We repeat the linear regression experiments from Section 4 but add progressively more Gaussian noise ($\mu = 0$, $\sigma \in \{0, 0.1, 0.5, 1\}$) to the outputs of the in-context input-output pairs. As expected, performance degrades with increasing noise. However, the relative performance differences among the architectures remain largely unchanged. Results are shown in Figure 6.

## C Supplementary data for Section 5: image classification

We show examples of the sequences used for training and evaluation in Figure 7. The single training run with achieved the best validation accuracy is shown in Figure 8 as as line plot. Tabular views of the experiments in this section are shown in Table 13 and 14.

## D Supplementary data for Section 6: Language Modeling

Normalized accuracies for the simple in-context learning experiment are shown in Figure 9.

## E Transformer Positional Embedding Abalations

Given the poor extrapolation abilities observed in transformers, we decided to test the effects of various positional embeddings, namely: sinusoidal (Vaswani et al., 2017), learned absolute (Radford et al., 2019), rotary (Su et al., 2022), and ALiBi (Press et al., 2022). We also tested the effects of removing positional embeddings entirely. To ensure that each transformer variant is identical in design (except for positional embedding), we use the `x-transformers` library.

Associative recall is shown in Table 15. We observe that performance for prompt lengths seen during training are nearly identical across positional embeddings. However, when considering the best run per model, only sinusoidal and no positional embeddings extrapolate well, reaching and maintaining near perfect accuracy across prompt lengths when $|V| = 40$. On average (across training runs), sinusoidal and no embeddings still extrapolate better than other options but do not always reach and maintain perfect accuracy.

Linear regression is shown in Table 16. Again, performance for prompt lengths seen during training are nearly identical across embedding options. While no training run demonstrated consistency, removing positional embeddings extrapolated better than all other options.

Multiclass classification is shown in Table 17. Performance, again was nearly identical for prompt lenghts seen during training. Differences in extrapolation ability were less pronounced for this task but removing positional embeddings was still the top performer on average.

Language modeling is shown in Table 10. While ALiBi did not extrapolate well in the previous experiments, we found that it resulted in the best validation loss for language modeling, followed

by rotary embeddings. Removing positional embeddings resulted in the worst language modeling validation loss.

## F    PERMUTATION INVARIANCE EXPERIMENTS

This experiment measures the effects of positional embeddings given that in-context examples in our tasks should be permutation invariant. Results are shown in Figure 11.

Specifically, we consider the following variables:

Token representation scheme: We represent in-context example pairs as a single token (instead of two in our original experiments) which allows us to remove positional embeddings. Specifically, we either sum or concatenate their embeddings. The query label is masked out by setting its embedding to zero.

Positional embeddings: whether to use learned absolute positional embeddings or no positional embeddings at all.

Attention mask: encoder-only vs decoder-only transformer. Note that in both scenarios, the query can attend to all in-context examples. In the encoder-only transformer, each example can attend to all other examples since it does not employ a causal mask. Examples in the decoder-only transformer can only attend to examples to its left.

The remaining settings are identical to Section 4 with the following changes: Our hyperparameter sweep covers 2 learning rates, 2 seeds, and 2 layer depths. We train for 50K steps and only take the loss (and evaluate) at the token index 32 (i.e., models are trained to make a single prediction given 31 example pairs and the query). We conducted 768 training runs in total.

We make the following observations:

Token representation scheme sensitivity: Associative recall and multiclass classification are not sensitive to tokenization schemes. However, we observe that concatenating embeddings in linear regression and image classification resulted in noticeably improved performance. We suspect that it is easier for attention heads to discern in-context inputs from outputs if they initially reside in their own subspace. Removing positional embeddings did not impact performance. This makes intuitive sense as in-context examples in this setting are permutation invariant. For most tasks, encoder-only and decoder-only transformers perform on par. The exception was linear regression where the encoder-only outperformed the decoder-only in the more difficult settings (d=20, 30). For image classification, we observed that ICL emerged in both transformers in very similar windows and followed a similar decay scheduled (as discussed in Section 5).

Table 2: Hyperparameters for linear regression, multiclass classification, associative recall, and image classification experiments.

| | |
|---|---|
| Optimizer | AdamW |
| $\beta_1, \beta_2$ | 0.90, 0.95 |
| Learning rate | {1e-3, 3e-4, 1e-4} |
| Warmup schedule | linear |
| Learning rate schedule | cosine decay |
| Training iterations | 100,000 |
| Batch size | 128 |
| Layers | {4, 8, 12} |
| Embedding size | 64 |
| Seed | {8, 16, 32} |

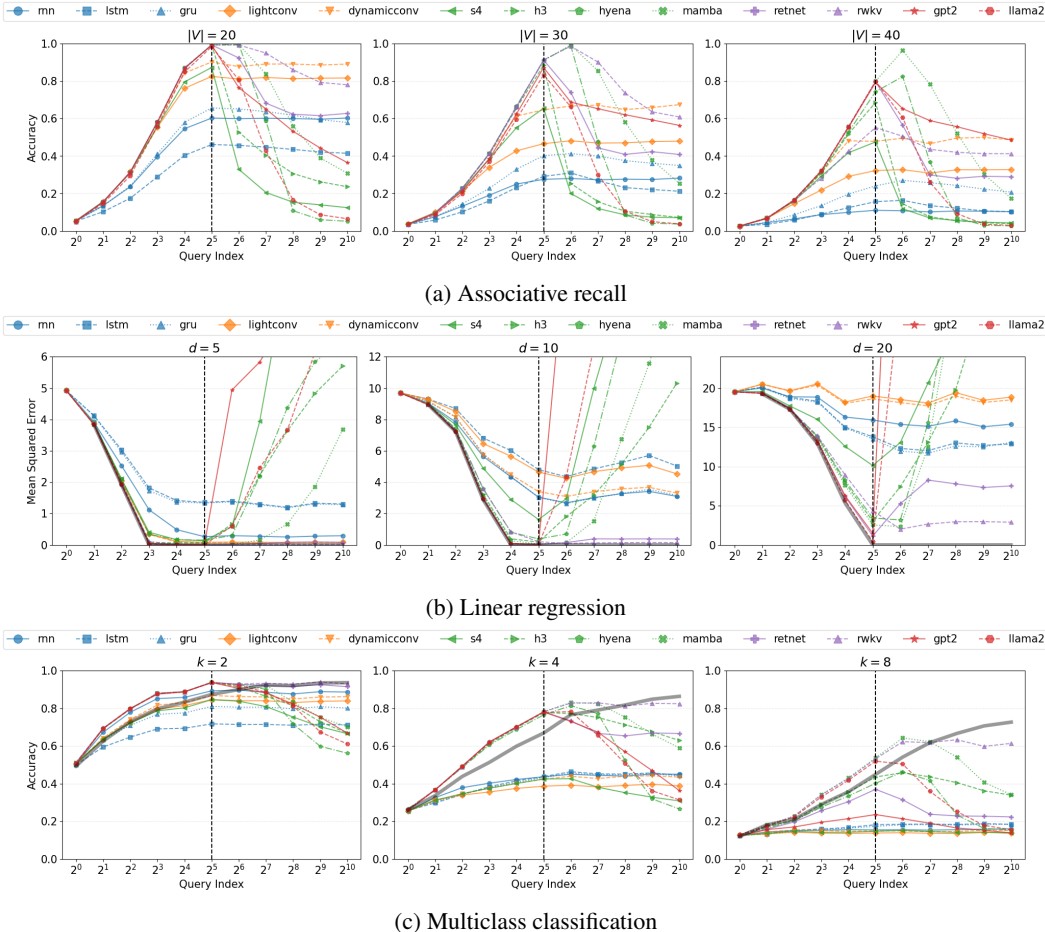

Figure 5: *Evaluating various architectures on in-context learning associative recall, linear regression, and multiclass classification.* We plot average test accuracy and mean squared error as a function of the number of in-context examples. A query index of $2^5 = 32$ implies 31 in-context examples, which is also the highest number of in-context examples seen during training (vertical dotted line). Task difficulty increases from left to right. Each line represents an average over all training runs for a given combination of task, difficulty, and architecture. Classical baselines (black) are shown for linear regression (ridge regression) and multiclass classification (logistic regression). See Tables 10, 8, 12 for a tabular view of the same data. See Figure 1 for the training runs that achieved the best performance.

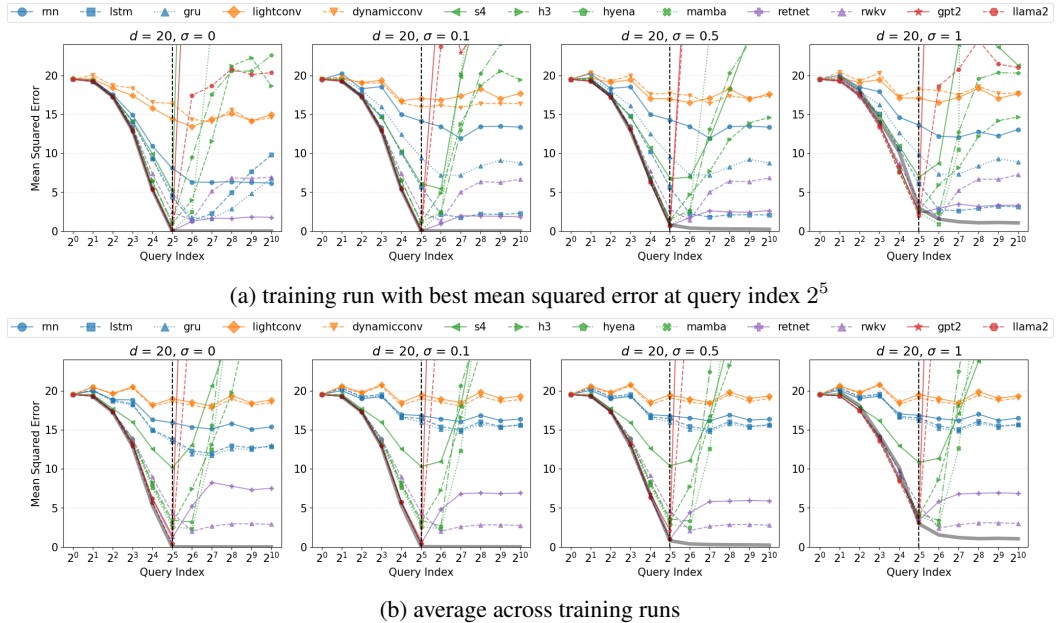

Figure 6: *Linear regression with Gaussian noise.* We plot mean squared error as a function of the number of in-context examples. Ridge regression is shown in black.

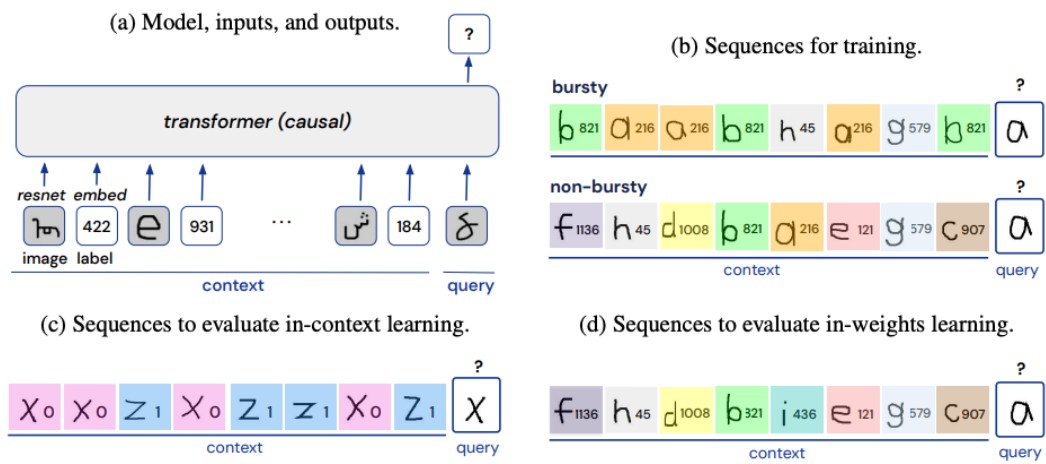

Figure 7: Image classification experimental design as outlined in Section 5. Figure taken from Chan et al. (2022) and included here for the reader's convenience. **(a)** "transformer" can be replaced with any of our architectures, e.g., RWKV. **(d)** This subplot can be safely ignored because we do not evaluate in-weights learning.

Table 3: Embedding sizes and layers for normalizing parameters to approximately 500K in linear regression, multiclass classification, associative recall, and image classification experiments.

|  | Layers | Embedding Size |
|---|---|---|
| S4 | 5 | 96 |
| DYNAMICCONV | 5 | 96 |
| LSTM | 4 | 128 |
| LIGHTCONV | 5 | 96 |
| GRU | 5 | 128 |
| RNN | 5 | 224 |

Table 4: Hyperparameters for language modeling experiments.

| Optimizer | AdamW |
|---|---|
| $\beta_1, \beta_2$ | 0.90, 0.95 |
| Learning rate | 3e-4 |
| Warmup schedule | linear |
| Learning rate schedule | cosine decay |
| Training iterations | 200,000 |
| Batch size | 50 |
| Sequence length | 512 |
| Layers | 8 |
| Embedding size | 512 |

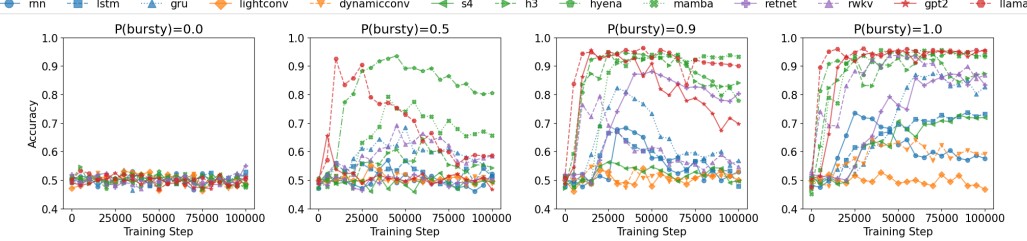

Figure 8: *Measuring the effects training data distributional properties on in-context learning.* We plot test accuracy as a function of training steps. P(bursty) indicates the proportion of training samples that were bursty. The remaining samples are non-bursty (i.i.d in-context examples). Each line represents the single run that achieved the best validation accuracy. See Table 13 for a tabular view of the same data. See Figure 2 for average test accuracy (across runs).

Table 5: Embedding sizes and layers for normalizing parameters to approximately 33M in language modeling experiments.

|  | Layers | Embedding Size |
|---|---|---|
| GPT2 | 8 | 576 |
| RWKV | 6 | 640 |
| HYENA | 8 | 576 |
| H3 | 7 | 1024 |
| S4 | 6 | 768 |
| DYNAMICCONV | 7 | 640 |
| LSTM | 5 | 896 |
| LIGHTCONV | 5 | 768 |
| GRU | 5 | 1024 |
| RNN | 5 | 1792 |

Table 6: Summary statistics for TinyStories dataset used for language modeling.

|  | Training | Validation |
|---|---|---|
| Total stories | 2,119,719 | 21,990 |
| Total tokens | 512,274,933 | 5,151,931 |
| Unique tokens | 15,200 | 8,235 |
| Average tokens | 241 | 234 |
| Median tokens | 208 | 205 |
| Standard deviation | 116 | 109 |
| Shortest story | 0 | 17 |
| Longest story | 1,431 | 1,183 |

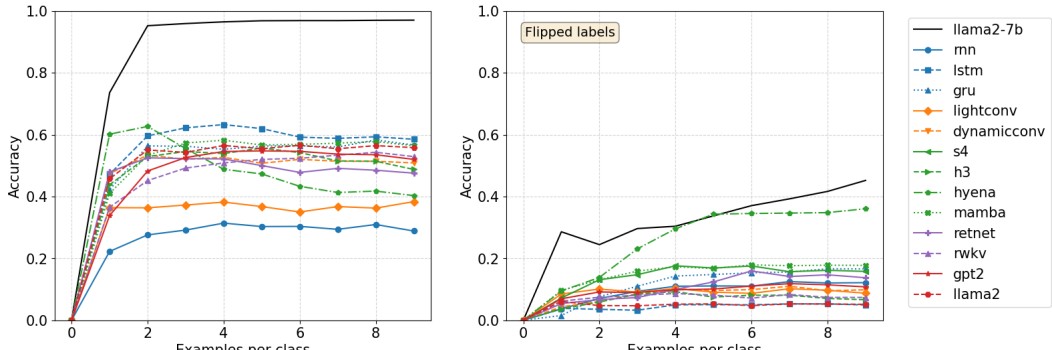

Figure 9: *Evaluating various architectures on a simple natural language ICL task.* We report accuracy as a function of the number of in-context examples. Accuracy is normalized with respect to accuracy when given 0 examples. We use the open sourced weights for Llama2-7B and do not fine-tune. All other models are trained from scratch and are no larger than 33M parameters (excluding embedding layers). **Right:** Flipped label setting, i.e., "happy" is replaced with "sad" and vice versa. See Figure 4 for unnormalized accuracy.

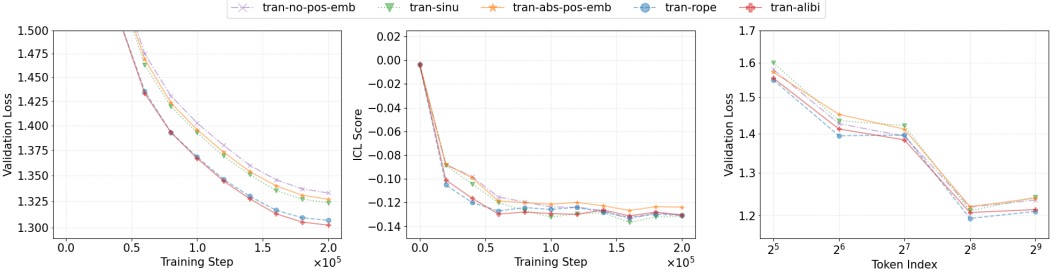

Figure 10: Language modeling experiments repeated across various transformer positional embedding options.

Query Index

| | $2^0$ | $2^1$ | $2^2$ | $2^3$ | $2^4$ | $2^5$ | $2^6$ | $2^7$ | $2^8$ | $2^9$ | $2^{10}$ |
|---|---|---|---|---|---|---|---|---|---|---|---|
| mn | 0.05 | 0.16 | 0.33 | 0.59 | 0.86 | 0.98 | 0.97 | 0.97 | 0.98 | 0.98 | 0.98 |
| lstm | 0.05 | 0.16 | 0.32 | 0.59 | 0.87 | 0.99 | 1.0 | 1.0 | 1.0 | 0.99 | 0.99 |
| gru | 0.06 | 0.16 | 0.31 | 0.58 | 0.87 | 0.99 | 0.99 | 0.98 | 0.86 | 0.69 | 0.51 |
| lightconv | 0.06 | 0.16 | 0.33 | 0.58 | 0.86 | 0.96 | 0.97 | 0.97 | 0.97 | 0.96 | 0.97 |
| dynamicconv | 0.05 | 0.15 | 0.32 | 0.58 | 0.87 | 0.98 | 0.97 | 0.98 | 0.98 | 0.98 | 0.98 |
| s4 | 0.05 | 0.14 | 0.33 | 0.58 | 0.87 | 0.98 | 0.53 | 0.33 | 0.22 | 0.15 | 0.15 |
| h3 | 0.05 | 0.16 | 0.32 | 0.58 | 0.86 | 0.99 | 0.46 | 0.29 | 0.22 | 0.18 | 0.14 |
| hyena | 0.05 | 0.16 | 0.31 | 0.59 | 0.87 | 1.0 | 1.0 | 0.22 | 0.06 | 0.05 | 0.04 |
| mamba | 0.05 | 0.16 | 0.31 | 0.59 | 0.87 | 1.0 | 1.0 | 0.87 | 0.6 | 0.4 | 0.27 |
| retnet | 0.04 | 0.16 | 0.32 | 0.58 | 0.86 | 0.99 | 0.96 | 0.91 | 0.91 | 0.9 | 0.9 |
| rwkv | 0.04 | 0.16 | 0.31 | 0.58 | 0.87 | 1.0 | 0.99 | 0.94 | 0.87 | 0.78 | 0.8 |
| gpt2 | 0.06 | 0.16 | 0.32 | 0.58 | 0.86 | 1.0 | 0.76 | 0.58 | 0.38 | 0.1 | 0.09 |
| llama2 | 0.05 | 0.16 | 0.31 | 0.58 | 0.87 | 1.0 | 0.83 | 0.34 | 0.08 | 0.06 | 0.05 |

$|V| = 20$, best training run

Query Index

| | $2^0$ | $2^1$ | $2^2$ | $2^3$ | $2^4$ | $2^5$ | $2^6$ | $2^7$ | $2^8$ | $2^9$ | $2^{10}$ |
|---|---|---|---|---|---|---|---|---|---|---|---|
| mn | 0.04 | 0.1 | 0.23 | 0.41 | 0.65 | 0.76 | 0.78 | 0.77 | 0.76 | 0.77 | 0.78 |
| lstm | 0.03 | 0.1 | 0.23 | 0.42 | 0.66 | 0.91 | 0.96 | 0.71 | 0.56 | 0.51 | 0.49 |
| gru | 0.03 | 0.09 | 0.23 | 0.42 | 0.66 | 0.9 | 0.98 | 0.97 | 0.91 | 0.84 | 0.79 |
| lightconv | 0.05 | 0.09 | 0.22 | 0.41 | 0.64 | 0.76 | 0.76 | 0.76 | 0.76 | 0.75 | 0.74 |
| dynamicconv | 0.04 | 0.1 | 0.23 | 0.41 | 0.66 | 0.86 | 0.88 | 0.89 | 0.86 | 0.87 | 0.88 |
| s4 | 0.03 | 0.1 | 0.23 | 0.41 | 0.63 | 0.81 | 0.36 | 0.15 | 0.09 | 0.06 | 0.07 |
| h3 | 0.04 | 0.1 | 0.23 | 0.41 | 0.66 | 0.91 | 0.21 | 0.11 | 0.07 | 0.05 | 0.06 |
| hyena | 0.04 | 0.09 | 0.23 | 0.41 | 0.67 | 0.91 | 1.0 | 0.15 | 0.03 | 0.03 | 0.04 |
| mamba | 0.04 | 0.09 | 0.24 | 0.4 | 0.67 | 0.91 | 1.0 | 0.88 | 0.54 | 0.3 | 0.19 |
| retnet | 0.03 | 0.1 | 0.23 | 0.42 | 0.66 | 0.92 | 0.83 | 0.63 | 0.61 | 0.61 | 0.62 |
| rwkv | 0.04 | 0.09 | 0.23 | 0.42 | 0.67 | 0.92 | 1.0 | 0.87 | 0.67 | 0.47 | 0.4 |
| gpt2 | 0.04 | 0.09 | 0.23 | 0.42 | 0.67 | 0.91 | 0.9 | 0.88 | 0.88 | 0.85 | 0.78 |
| llama2 | 0.04 | 0.09 | 0.22 | 0.41 | 0.67 | 0.91 | 0.89 | 0.52 | 0.2 | 0.04 | 0.03 |

$|V| = 30$, best training run

Query Index

| | $2^0$ | $2^1$ | $2^2$ | $2^3$ | $2^4$ | $2^5$ | $2^6$ | $2^7$ | $2^8$ | $2^9$ | $2^{10}$ |
|---|---|---|---|---|---|---|---|---|---|---|---|
| mn | 0.02 | 0.07 | 0.16 | 0.28 | 0.35 | 0.4 | 0.41 | 0.35 | 0.4 | 0.39 | 0.38 |
| lstm | 0.03 | 0.07 | 0.17 | 0.32 | 0.53 | 0.74 | 0.77 | 0.55 | 0.35 | 0.25 | 0.24 |
| gru | 0.03 | 0.07 | 0.16 | 0.32 | 0.54 | 0.71 | 0.83 | 0.82 | 0.75 | 0.62 | 0.54 |
| lightconv | 0.03 | 0.07 | 0.17 | 0.32 | 0.49 | 0.53 | 0.54 | 0.53 | 0.53 | 0.56 | 0.57 |
| dynamicconv | 0.03 | 0.06 | 0.17 | 0.32 | 0.56 | 0.61 | 0.63 | 0.61 | 0.65 | 0.63 | 0.6 |
| s4 | 0.03 | 0.08 | 0.17 | 0.32 | 0.53 | 0.64 | 0.18 | 0.09 | 0.05 | 0.03 | 0.03 |
| h3 | 0.03 | 0.07 | 0.16 | 0.33 | 0.55 | 0.81 | 0.21 | 0.12 | 0.08 | 0.04 | 0.04 |
| hyena | 0.02 | 0.06 | 0.17 | 0.32 | 0.56 | 0.79 | 0.99 | 0.93 | 0.45 | 0.09 | 0.05 |
| mamba | 0.03 | 0.08 | 0.17 | 0.32 | 0.56 | 0.8 | 0.96 | 0.73 | 0.37 | 0.14 | 0.07 |
| retnet | 0.02 | 0.07 | 0.17 | 0.32 | 0.55 | 0.79 | 0.63 | 0.28 | 0.26 | 0.26 | 0.26 |
| rwkv | 0.02 | 0.07 | 0.16 | 0.32 | 0.56 | 0.8 | 0.99 | 0.99 | 0.72 | 0.67 | 0.68 |
| gpt2 | 0.02 | 0.08 | 0.17 | 0.32 | 0.55 | 0.8 | 0.3 | 0.2 | 0.16 | 0.11 | 0.09 |
| llama2 | 0.02 | 0.07 | 0.16 | 0.33 | 0.55 | 0.8 | 0.51 | 0.19 | 0.09 | 0.04 | 0.03 |

$|V| = 40$, best training run

Table 7: Associative recall best accuracy. See Figure 1 for line plots of the same data

## Query Index

| | $2^0$ | $2^1$ | $2^2$ | $2^3$ | $2^4$ | $2^5$ | $2^6$ | $2^7$ | $2^8$ | $2^9$ | $2^{10}$ |
|---|---|---|---|---|---|---|---|---|---|---|---|
| rnn | 0.05 | 0.13 | 0.24 | 0.4 | 0.55 | 0.6 | 0.6 | 0.6 | 0.6 | 0.6 | 0.6 |
| lstm | 0.05 | 0.1 | 0.17 | 0.29 | 0.4 | 0.46 | 0.46 | 0.45 | 0.44 | 0.42 | 0.41 |
| gru | 0.05 | 0.13 | 0.24 | 0.41 | 0.58 | 0.66 | 0.65 | 0.64 | 0.62 | 0.59 | 0.58 |
| lightconv | 0.05 | 0.16 | 0.32 | 0.55 | 0.76 | 0.83 | 0.81 | 0.82 | 0.81 | 0.82 | 0.82 |
| dynamicconv | 0.05 | 0.15 | 0.31 | 0.58 | 0.84 | 0.9 | 0.88 | 0.89 | 0.89 | 0.89 | 0.89 |
| s4 | 0.06 | 0.16 | 0.31 | 0.56 | 0.8 | 0.87 | 0.33 | 0.21 | 0.15 | 0.14 | 0.12 |
| h3 | 0.05 | 0.15 | 0.31 | 0.58 | 0.87 | 0.99 | 0.53 | 0.4 | 0.31 | 0.26 | 0.24 |
| hyena | 0.05 | 0.15 | 0.31 | 0.58 | 0.87 | 0.99 | 1.0 | 0.59 | 0.11 | 0.06 | 0.05 |
| mamba | 0.05 | 0.16 | 0.31 | 0.58 | 0.87 | 0.99 | 0.99 | 0.84 | 0.56 | 0.39 | 0.31 |
| retnet | 0.05 | 0.15 | 0.31 | 0.58 | 0.87 | 0.99 | 0.92 | 0.68 | 0.62 | 0.62 | 0.63 |
| rwkv | 0.05 | 0.16 | 0.31 | 0.58 | 0.87 | 0.99 | 0.99 | 0.95 | 0.86 | 0.79 | 0.78 |
| gpt2 | 0.05 | 0.15 | 0.31 | 0.58 | 0.87 | 0.99 | 0.77 | 0.65 | 0.53 | 0.44 | 0.36 |
| llama2 | 0.05 | 0.15 | 0.29 | 0.56 | 0.85 | 0.98 | 0.81 | 0.43 | 0.17 | 0.09 | 0.06 |

$|V| = 20$, average over all training runs

## Query Index

| | $2^0$ | $2^1$ | $2^2$ | $2^3$ | $2^4$ | $2^5$ | $2^6$ | $2^7$ | $2^8$ | $2^9$ | $2^{10}$ |
|---|---|---|---|---|---|---|---|---|---|---|---|
| rnn | 0.04 | 0.08 | 0.13 | 0.19 | 0.25 | 0.28 | 0.28 | 0.27 | 0.28 | 0.27 | 0.28 |
| lstm | 0.03 | 0.06 | 0.1 | 0.16 | 0.23 | 0.29 | 0.31 | 0.27 | 0.23 | 0.22 | 0.21 |
| gru | 0.04 | 0.07 | 0.14 | 0.23 | 0.33 | 0.4 | 0.41 | 0.4 | 0.38 | 0.36 | 0.35 |
| lightconv | 0.04 | 0.1 | 0.21 | 0.34 | 0.43 | 0.47 | 0.48 | 0.47 | 0.47 | 0.48 | 0.48 |
| dynamicconv | 0.03 | 0.1 | 0.22 | 0.41 | 0.61 | 0.65 | 0.67 | 0.67 | 0.65 | 0.66 | 0.67 |
| s4 | 0.04 | 0.1 | 0.22 | 0.39 | 0.55 | 0.66 | 0.2 | 0.12 | 0.09 | 0.07 | 0.07 |
| h3 | 0.03 | 0.1 | 0.23 | 0.41 | 0.66 | 0.89 | 0.25 | 0.16 | 0.11 | 0.09 | 0.07 |
| hyena | 0.04 | 0.1 | 0.23 | 0.41 | 0.67 | 0.91 | 0.99 | 0.48 | 0.09 | 0.04 | 0.04 |
| mamba | 0.04 | 0.1 | 0.23 | 0.41 | 0.66 | 0.91 | 0.99 | 0.86 | 0.58 | 0.38 | 0.25 |
| retnet | 0.04 | 0.1 | 0.23 | 0.41 | 0.66 | 0.91 | 0.74 | 0.44 | 0.41 | 0.42 | 0.41 |
| rwkv | 0.03 | 0.1 | 0.23 | 0.41 | 0.66 | 0.91 | 0.99 | 0.9 | 0.74 | 0.64 | 0.61 |
| gpt2 | 0.04 | 0.09 | 0.21 | 0.39 | 0.63 | 0.87 | 0.69 | 0.65 | 0.62 | 0.59 | 0.56 |
| llama2 | 0.04 | 0.09 | 0.2 | 0.37 | 0.6 | 0.83 | 0.66 | 0.3 | 0.11 | 0.05 | 0.04 |

$|V| = 30$, average over all training runs

## Query Index

| | $2^0$ | $2^1$ | $2^2$ | $2^3$ | $2^4$ | $2^5$ | $2^6$ | $2^7$ | $2^8$ | $2^9$ | $2^{10}$ |
|---|---|---|---|---|---|---|---|---|---|---|---|
| rnn | 0.03 | 0.04 | 0.06 | 0.09 | 0.1 | 0.11 | 0.11 | 0.1 | 0.11 | 0.11 | 0.1 |
| lstm | 0.03 | 0.04 | 0.06 | 0.09 | 0.12 | 0.16 | 0.16 | 0.14 | 0.12 | 0.11 | 0.1 |
| gru | 0.03 | 0.05 | 0.09 | 0.14 | 0.2 | 0.24 | 0.27 | 0.26 | 0.24 | 0.22 | 0.21 |
| lightconv | 0.02 | 0.07 | 0.15 | 0.22 | 0.29 | 0.32 | 0.33 | 0.31 | 0.33 | 0.33 | 0.33 |
| dynamicconv | 0.03 | 0.07 | 0.16 | 0.31 | 0.48 | 0.48 | 0.5 | 0.47 | 0.5 | 0.5 | 0.49 |
| s4 | 0.03 | 0.07 | 0.16 | 0.29 | 0.42 | 0.48 | 0.12 | 0.07 | 0.05 | 0.05 | 0.04 |
| h3 | 0.03 | 0.07 | 0.16 | 0.31 | 0.52 | 0.69 | 0.14 | 0.08 | 0.06 | 0.04 | 0.04 |
| hyena | 0.02 | 0.07 | 0.16 | 0.32 | 0.53 | 0.74 | 0.82 | 0.37 | 0.07 | 0.03 | 0.03 |
| mamba | 0.03 | 0.07 | 0.17 | 0.32 | 0.56 | 0.8 | 0.96 | 0.78 | 0.52 | 0.3 | 0.17 |
| retnet | 0.02 | 0.07 | 0.17 | 0.32 | 0.56 | 0.8 | 0.56 | 0.3 | 0.28 | 0.29 | 0.29 |
| rwkv | 0.03 | 0.07 | 0.16 | 0.28 | 0.43 | 0.55 | 0.51 | 0.44 | 0.42 | 0.41 | 0.41 |
| gpt2 | 0.03 | 0.07 | 0.16 | 0.32 | 0.55 | 0.79 | 0.65 | 0.59 | 0.56 | 0.52 | 0.49 |
| llama2 | 0.03 | 0.07 | 0.16 | 0.32 | 0.56 | 0.8 | 0.61 | 0.26 | 0.09 | 0.04 | 0.03 |

$|V| = 40$, average over all training runs

Table 8: Associative recall average accuracy. See Figure 5 for line plots of the same data.

Query Index

| | $2^0$ | $2^1$ | $2^2$ | $2^3$ | $2^4$ | $2^5$ | $2^6$ | $2^7$ | $2^8$ | $2^9$ | $2^{10}$ |
|---|---|---|---|---|---|---|---|---|---|---|---|
| mn | 4.93 | 3.86 | 1.92 | 0.26 | 0.04 | 0.02 | 0.03 | 0.02 | 0.02 | 0.03 | 0.02 |
| lstm | 4.93 | 3.82 | 1.95 | 0.07 | 0.0 | 0.0 | 0.0 | 0.0 | 0.0 | 0.0 | 0.0 |
| gru | 4.93 | 3.83 | 1.98 | 0.04 | 0.0 | 0.0 | 0.0 | 0.0 | 0.0 | 0.0 | 0.0 |
| lightconv | 4.93 | 3.84 | 1.91 | 0.13 | 0.01 | 0.01 | 0.01 | 0.01 | 0.01 | 0.01 | 0.01 |
| dynamicconv | 4.93 | 3.83 | 1.93 | 0.06 | 0.01 | 0.01 | 0.01 | 0.01 | 0.01 | 0.01 | 0.01 |
| s4 | 4.93 | 3.86 | 2.04 | 0.17 | 0.07 | 0.05 | 0.21 | 0.52 | 2.68 | 3.68 | 5.19 |
| h3 | 4.93 | 3.85 | 1.97 | 0.06 | 0.05 | 0.05 | 0.45 | 0.78 | 0.99 | 1.27 | 1.98 |
| hyena | 4.92 | 3.87 | 2.05 | 0.16 | 0.08 | 0.07 | 0.1 | 3.02 | 4.77 | 5.18 | 5.47 |
| mamba | 4.93 | 3.84 | 1.93 | 0.01 | 0.0 | 0.0 | 0.0 | 0.01 | 0.04 | 0.12 | 0.31 |
| retnet | 4.93 | 3.84 | 1.93 | 0.0 | 0.0 | 0.0 | 0.0 | 0.0 | 0.0 | 0.0 | 0.0 |
| rwkv | 4.93 | 3.85 | 1.94 | 0.02 | 0.0 | 0.0 | 0.0 | 0.0 | 0.0 | 0.0 | 0.0 |
| gpt2 | 4.93 | 3.84 | 1.92 | 0.0 | 0.0 | 0.0 | 3.2 | 3.76 | 4.12 | 5.72 | 5.97 |
| llama2 | 4.93 | 3.85 | 1.92 | 0.0 | 0.0 | 0.0 | 0.0 | 0.03 | 0.09 | 1.2 | 4.19 |

$d = 5$, best training run

Query Index

| | $2^0$ | $2^1$ | $2^2$ | $2^3$ | $2^4$ | $2^5$ | $2^6$ | $2^7$ | $2^8$ | $2^9$ | $2^{10}$ |
|---|---|---|---|---|---|---|---|---|---|---|---|
| mn | 9.7 | 9.05 | 7.46 | 4.37 | 1.96 | 0.6 | 0.34 | 0.38 | 0.43 | 0.5 | 0.44 |
| lstm | 9.69 | 9.02 | 7.23 | 3.9 | 0.95 | 0.17 | 0.1 | 0.2 | 0.45 | 0.55 | 0.77 |
| gru | 9.69 | 8.99 | 7.31 | 3.37 | 0.49 | 0.08 | 0.06 | 0.1 | 0.18 | 0.39 | 0.56 |
| lightconv | 9.69 | 9.02 | 7.41 | 4.41 | 1.85 | 0.73 | 0.64 | 0.66 | 0.69 | 0.76 | 0.69 |
| dynamicconv | 9.69 | 8.99 | 7.37 | 4.09 | 1.51 | 0.43 | 0.28 | 0.31 | 0.32 | 0.33 | 0.33 |
| s4 | 9.7 | 8.96 | 7.3 | 4.13 | 1.48 | 0.43 | 2.02 | 16.72 | 16.04 | 13.48 | 10.91 |
| h3 | 9.69 | 8.92 | 7.27 | 3.01 | 0.16 | 0.09 | 1.26 | 1.52 | 2.25 | 3.84 | 6.85 |
| hyena | 9.69 | 8.94 | 7.23 | 3.02 | 0.17 | 0.14 | 2.37 | 8.39 | 10.55 | 11.37 | 9.94 |
| mamba | 9.69 | 8.96 | 7.23 | 2.91 | 0.03 | 0.01 | 0.03 | 0.47 | 1.39 | 3.12 | 4.61 |
| retnet | 9.69 | 8.97 | 7.21 | 2.88 | 0.0 | 0.0 | 0.08 | 0.12 | 0.11 | 0.11 | 0.1 |
| rwkv | 9.69 | 9.01 | 7.17 | 2.99 | 0.1 | 0.02 | 0.05 | 0.14 | 0.2 | 0.27 | 0.26 |
| gpt2 | 9.69 | 8.95 | 7.23 | 2.9 | 0.01 | 0.0 | 11.59 | 14.45 | 19.56 | 25.99 | 25.85 |
| llama2 | 9.69 | 8.97 | 7.22 | 2.88 | 0.0 | 0.0 | 0.39 | 2.48 | 4.08 | 7.96 | 9.26 |

$d = 10$, best training run

Query Index

| | $2^0$ | $2^1$ | $2^2$ | $2^3$ | $2^4$ | $2^5$ | $2^6$ | $2^7$ | $2^8$ | $2^9$ | $2^{10}$ |
|---|---|---|---|---|---|---|---|---|---|---|---|
| mn | 19.65 | 19.44 | 17.6 | 14.94 | 10.96 | 8.08 | 6.3 | 6.28 | 6.34 | 6.25 | 6.17 |
| lstm | 19.52 | 19.21 | 17.36 | 14.08 | 9.33 | 4.39 | 1.34 | 2.25 | 4.97 | 7.68 | 9.8 |
| gru | 19.53 | 19.37 | 17.3 | 14.1 | 9.43 | 4.94 | 1.72 | 1.61 | 2.88 | 4.85 | 6.78 |
| lightconv | 19.63 | 19.55 | 18.41 | 17.46 | 15.83 | 14.43 | 13.44 | 14.42 | 15.12 | 14.19 | 15.0 |
| dynamicconv | 19.5 | 20.08 | 18.76 | 18.4 | 16.58 | 16.41 | 14.39 | 14.04 | 15.63 | 14.18 | 14.59 |
| s4 | 19.52 | 19.33 | 17.32 | 14.19 | 9.94 | 5.23 | 87.99 | 121.33 | 71.66 | 83.56 | 90.5 |
| h3 | 19.52 | 19.37 | 17.32 | 13.4 | 6.39 | 0.96 | 4.0 | 11.63 | 21.26 | 22.31 | 18.68 |
| hyena | 19.52 | 19.33 | 17.34 | 13.31 | 6.49 | 1.16 | 9.5 | 17.6 | 20.57 | 20.6 | 22.66 |
| mamba | 19.52 | 19.27 | 17.28 | 12.94 | 5.54 | 0.03 | 2.46 | 26.73 | 32.55 | 38.58 | 41.25 |
| retnet | 19.53 | 19.27 | 17.29 | 12.95 | 5.39 | 0.02 | 1.24 | 1.65 | 1.62 | 1.81 | 1.77 |
| rwkv | 19.52 | 19.34 | 17.41 | 13.51 | 7.39 | 2.44 | 1.38 | 5.15 | 6.84 | 6.78 | 6.94 |
| gpt2 | 19.52 | 19.28 | 17.26 | 12.98 | 5.44 | 0.04 | 56.04 | 74.17 | 79.48 | 83.38 | 81.58 |
| llama2 | 19.52 | 19.3 | 17.33 | 12.94 | 5.37 | 0.01 | 17.47 | 18.69 | 20.87 | 20.14 | 20.4 |

$d = 20$, best training run

Table 9: Linear regression best mean squared error. See Figure 1 for line plots of the same data

Query Index

| | $2^0$ | $2^1$ | $2^2$ | $2^3$ | $2^4$ | $2^5$ | $2^6$ | $2^7$ | $2^8$ | $2^9$ | $2^{10}$ |
|---|---|---|---|---|---|---|---|---|---|---|---|
| mn | 4.94 | 3.93 | 2.53 | 1.12 | 0.49 | 0.26 | 0.29 | 0.28 | 0.25 | 0.28 | 0.29 |
| lstm | 4.93 | 4.13 | 3.04 | 1.82 | 1.42 | 1.36 | 1.39 | 1.3 | 1.2 | 1.33 | 1.3 |
| gru | 4.93 | 4.12 | 2.97 | 1.72 | 1.37 | 1.34 | 1.37 | 1.28 | 1.18 | 1.31 | 1.28 |
| lightconv | 4.93 | 3.87 | 2.06 | 0.35 | 0.11 | 0.08 | 0.09 | 0.08 | 0.08 | 0.09 | 0.08 |
| dynamicconv | 4.93 | 3.85 | 2.04 | 0.33 | 0.09 | 0.06 | 0.07 | 0.07 | 0.06 | 0.07 | 0.07 |
| s4 | 4.93 | 3.88 | 2.11 | 0.4 | 0.16 | 0.13 | 0.58 | 3.94 | 8.85 | 8.52 | 8.46 |
| h3 | 4.93 | 3.85 | 1.98 | 0.09 | 0.06 | 0.06 | 0.66 | 2.21 | 3.64 | 4.83 | 5.7 |
| hyena | 4.93 | 3.88 | 2.11 | 0.34 | 0.17 | 0.15 | 0.3 | 2.18 | 4.38 | 5.84 | 6.5 |
| mamba | 4.93 | 3.85 | 1.94 | 0.06 | 0.01 | 0.0 | 0.02 | 0.15 | 0.66 | 1.86 | 3.68 |
| retnet | 4.93 | 3.85 | 1.93 | 0.02 | 0.0 | 0.0 | 0.01 | 0.04 | 0.08 | 0.09 | 0.08 |
| rwkv | 4.93 | 3.86 | 1.96 | 0.1 | 0.01 | 0.01 | 0.0 | 0.0 | 0.0 | 0.01 | 0.01 |
| gpt2 | 4.93 | 3.85 | 1.93 | 0.05 | 0.0 | 0.0 | 4.94 | 5.83 | 7.28 | 8.85 | 9.31 |
| llama2 | 4.93 | 3.85 | 1.93 | 0.03 | 0.0 | 0.0 | 0.6 | 2.47 | 3.66 | 6.14 | 6.13 |

$d = 5$, average over all training runs

Query Index

| | $2^0$ | $2^1$ | $2^2$ | $2^3$ | $2^4$ | $2^5$ | $2^6$ | $2^7$ | $2^8$ | $2^9$ | $2^{10}$ |
|---|---|---|---|---|---|---|---|---|---|---|---|
| mn | 9.73 | 9.05 | 7.84 | 5.64 | 4.34 | 3.04 | 2.68 | 3.01 | 3.27 | 3.41 | 3.1 |
| lstm | 9.69 | 9.3 | 8.7 | 6.82 | 6.01 | 4.8 | 4.35 | 4.85 | 5.25 | 5.7 | 5.03 |
| gru | 9.69 | 9.14 | 8.17 | 5.81 | 4.37 | 3.03 | 2.64 | 2.98 | 3.28 | 3.54 | 3.16 |
| lightconv | 9.7 | 9.3 | 8.52 | 6.46 | 5.65 | 4.65 | 4.25 | 4.67 | 4.91 | 5.08 | 4.53 |
| dynamicconv | 9.7 | 9.19 | 8.16 | 5.74 | 4.49 | 3.4 | 3.09 | 3.4 | 3.57 | 3.67 | 3.31 |
| s4 | 9.69 | 9.02 | 7.67 | 4.89 | 2.9 | 1.62 | 2.95 | 9.99 | 16.05 | 18.91 | 19.55 |
| h3 | 9.69 | 8.94 | 7.29 | 3.23 | 0.39 | 0.17 | 1.8 | 3.18 | 5.15 | 7.51 | 10.32 |
| hyena | 9.69 | 8.99 | 7.43 | 3.58 | 0.81 | 0.38 | 0.7 | 6.28 | 13.23 | 14.44 | 19.46 |
| mamba | 9.69 | 8.98 | 7.25 | 3.16 | 0.27 | 0.05 | 0.14 | 1.52 | 6.74 | 11.57 | 17.21 |
| retnet | 9.69 | 8.97 | 7.23 | 2.9 | 0.04 | 0.01 | 0.15 | 0.39 | 0.38 | 0.38 | 0.38 |
| rwkv | 9.69 | 8.98 | 7.31 | 3.58 | 0.83 | 0.2 | 0.09 | 0.13 | 0.14 | 0.16 | 0.14 |
| gpt2 | 9.69 | 8.96 | 7.24 | 2.94 | 0.08 | 0.02 | 19.28 | 21.66 | 22.36 | 25.43 | 24.28 |
| llama2 | 9.69 | 8.96 | 7.23 | 2.95 | 0.1 | 0.02 | 4.33 | 12.59 | 16.89 | 19.21 | 17.83 |

$d = 10$, average over all training runs

Query Index

| | $2^0$ | $2^1$ | $2^2$ | $2^3$ | $2^4$ | $2^5$ | $2^6$ | $2^7$ | $2^8$ | $2^9$ | $2^{10}$ |
|---|---|---|---|---|---|---|---|---|---|---|---|
| mn | 19.58 | 20.02 | 18.89 | 18.86 | 16.31 | 15.94 | 15.38 | 15.13 | 15.82 | 15.08 | 15.39 |
| lstm | 19.52 | 20.14 | 18.82 | 18.34 | 15.03 | 13.79 | 12.31 | 12.02 | 13.01 | 12.69 | 12.88 |
| gru | 19.52 | 20.08 | 18.67 | 18.22 | 14.93 | 13.52 | 11.95 | 11.71 | 12.6 | 12.5 | 13.07 |
| lightconv | 19.53 | 20.5 | 19.66 | 20.55 | 18.21 | 19.0 | 18.55 | 18.08 | 19.43 | 18.45 | 18.85 |
| dynamicconv | 19.52 | 20.51 | 19.63 | 20.39 | 18.05 | 18.63 | 18.15 | 17.74 | 18.98 | 18.16 | 18.5 |
| s4 | 19.54 | 19.57 | 17.74 | 16.02 | 12.58 | 10.12 | 13.07 | 20.66 | 27.8 | 32.22 | 45.0 |
| h3 | 19.52 | 19.36 | 17.36 | 13.77 | 7.98 | 3.01 | 7.44 | 13.1 | 19.79 | 31.34 | 41.04 |
| hyena | 19.52 | 19.4 | 17.38 | 13.85 | 8.25 | 3.44 | 3.18 | 15.61 | 37.28 | 39.69 | 37.55 |
| mamba | 19.53 | 19.32 | 17.31 | 13.47 | 7.61 | 2.55 | 2.33 | 12.52 | 33.08 | 45.98 | 56.61 |
| retnet | 19.53 | 19.33 | 17.42 | 13.31 | 6.17 | 1.13 | 5.25 | 8.27 | 7.79 | 7.32 | 7.52 |
| rwkv | 19.53 | 19.32 | 17.34 | 13.94 | 8.94 | 4.37 | 2.01 | 2.66 | 2.95 | 2.96 | 2.89 |
| gpt2 | 19.52 | 19.37 | 17.4 | 13.3 | 6.28 | 1.48 | 77.25 | 69.72 | 64.41 | 62.52 | 60.67 |
| llama2 | 19.52 | 19.34 | 17.29 | 12.98 | 5.72 | 0.42 | 32.47 | 57.68 | 61.33 | 52.89 | 49.03 |

$d = 20$, average over all training runs

Table 10: Linear regression average mean squared error. See Figure 5 for line plots of the same data.

Query Index

| | $2^0$ | $2^1$ | $2^2$ | $2^3$ | $2^4$ | $2^5$ | $2^6$ | $2^7$ | $2^8$ | $2^9$ | $2^{10}$ |
|---|---|---|---|---|---|---|---|---|---|---|---|
| mn | 0.48 | 0.7 | 0.8 | 0.88 | 0.89 | 0.93 | 0.93 | 0.92 | 0.91 | 0.93 | 0.91 |
| lstm | 0.51 | 0.69 | 0.8 | 0.88 | 0.89 | 0.94 | 0.93 | 0.94 | 0.93 | 0.94 | 0.93 |
| gru | 0.5 | 0.69 | 0.8 | 0.87 | 0.88 | 0.94 | 0.93 | 0.93 | 0.91 | 0.93 | 0.91 |
| lightconv | 0.5 | 0.7 | 0.81 | 0.88 | 0.89 | 0.93 | 0.93 | 0.92 | 0.91 | 0.92 | 0.92 |
| dynamicconv | 0.53 | 0.69 | 0.8 | 0.87 | 0.89 | 0.93 | 0.91 | 0.9 | 0.9 | 0.92 | 0.91 |
| s4 | 0.52 | 0.7 | 0.8 | 0.88 | 0.9 | 0.94 | 0.93 | 0.92 | 0.9 | 0.88 | 0.85 |
| h3 | 0.51 | 0.69 | 0.8 | 0.88 | 0.89 | 0.94 | 0.91 | 0.89 | 0.89 | 0.66 | 0.19 |
| hyena | 0.5 | 0.69 | 0.8 | 0.88 | 0.89 | 0.94 | 0.92 | 0.93 | 0.66 | 0.5 | 0.51 |
| mamba | 0.51 | 0.69 | 0.79 | 0.89 | 0.89 | 0.94 | 0.93 | 0.93 | 0.91 | 0.9 | 0.89 |
| retnet | 0.49 | 0.7 | 0.8 | 0.88 | 0.89 | 0.93 | 0.92 | 0.9 | 0.88 | 0.91 | 0.9 |
| rwkv | 0.53 | 0.7 | 0.8 | 0.88 | 0.89 | 0.94 | 0.93 | 0.94 | 0.93 | 0.94 | 0.93 |
| gpt2 | 0.51 | 0.7 | 0.8 | 0.88 | 0.89 | 0.94 | 0.92 | 0.9 | 0.82 | 0.7 | 0.6 |
| llama2 | 0.49 | 0.69 | 0.8 | 0.87 | 0.89 | 0.93 | 0.93 | 0.91 | 0.88 | 0.74 | 0.63 |

$k = 2$, best training run

Query Index

| | $2^0$ | $2^1$ | $2^2$ | $2^3$ | $2^4$ | $2^5$ | $2^6$ | $2^7$ | $2^8$ | $2^9$ | $2^{10}$ |
|---|---|---|---|---|---|---|---|---|---|---|---|
| mn | 0.24 | 0.37 | 0.48 | 0.56 | 0.61 | 0.65 | 0.68 | 0.68 | 0.66 | 0.7 | 0.65 |
| lstm | 0.27 | 0.36 | 0.48 | 0.62 | 0.69 | 0.78 | 0.83 | 0.83 | 0.8 | 0.82 | 0.81 |
| gru | 0.27 | 0.37 | 0.49 | 0.63 | 0.7 | 0.78 | 0.83 | 0.84 | 0.83 | 0.84 | 0.85 |
| lightconv | 0.25 | 0.38 | 0.42 | 0.49 | 0.53 | 0.6 | 0.6 | 0.61 | 0.62 | 0.62 | 0.62 |
| dynamicconv | 0.24 | 0.36 | 0.44 | 0.49 | 0.54 | 0.59 | 0.61 | 0.6 | 0.6 | 0.62 | 0.6 |
| s4 | 0.24 | 0.32 | 0.42 | 0.47 | 0.48 | 0.56 | 0.53 | 0.3 | 0.25 | 0.26 | 0.23 |
| h3 | 0.27 | 0.36 | 0.51 | 0.61 | 0.71 | 0.78 | 0.75 | 0.75 | 0.69 | 0.7 | 0.63 |
| hyena | 0.27 | 0.35 | 0.5 | 0.61 | 0.7 | 0.78 | 0.83 | 0.83 | 0.61 | 0.29 | 0.23 |
| mamba | 0.24 | 0.35 | 0.49 | 0.62 | 0.7 | 0.77 | 0.83 | 0.83 | 0.71 | 0.56 | 0.46 |
| retnet | 0.27 | 0.37 | 0.49 | 0.63 | 0.7 | 0.78 | 0.81 | 0.79 | 0.77 | 0.78 | 0.78 |
| rwkv | 0.28 | 0.37 | 0.48 | 0.62 | 0.7 | 0.78 | 0.82 | 0.83 | 0.82 | 0.83 | 0.83 |
| gpt2 | 0.24 | 0.38 | 0.48 | 0.63 | 0.7 | 0.78 | 0.77 | 0.74 | 0.7 | 0.58 | 0.43 |
| llama2 | 0.25 | 0.38 | 0.48 | 0.61 | 0.71 | 0.78 | 0.8 | 0.67 | 0.54 | 0.34 | 0.32 |

$k = 4$, best training run

Query Index

| | $2^0$ | $2^1$ | $2^2$ | $2^3$ | $2^4$ | $2^5$ | $2^6$ | $2^7$ | $2^8$ | $2^9$ | $2^{10}$ |
|---|---|---|---|---|---|---|---|---|---|---|---|
| mn | 0.13 | 0.17 | 0.18 | 0.21 | 0.23 | 0.23 | 0.22 | 0.22 | 0.25 | 0.24 | 0.22 |
| lstm | 0.13 | 0.17 | 0.22 | 0.29 | 0.33 | 0.42 | 0.5 | 0.52 | 0.5 | 0.51 | 0.48 |
| gru | 0.12 | 0.15 | 0.19 | 0.22 | 0.25 | 0.24 | 0.28 | 0.28 | 0.28 | 0.27 | 0.28 |
| lightconv | 0.12 | 0.17 | 0.19 | 0.22 | 0.17 | 0.2 | 0.19 | 0.22 | 0.24 | 0.2 | 0.2 |
| dynamicconv | 0.15 | 0.19 | 0.22 | 0.26 | 0.24 | 0.25 | 0.24 | 0.25 | 0.25 | 0.23 | 0.26 |
| s4 | 0.13 | 0.18 | 0.18 | 0.22 | 0.2 | 0.2 | 0.22 | 0.19 | 0.2 | 0.18 | 0.19 |
| h3 | 0.12 | 0.18 | 0.24 | 0.33 | 0.44 | 0.53 | 0.6 | 0.6 | 0.55 | 0.52 | 0.48 |
| hyena | 0.14 | 0.2 | 0.24 | 0.35 | 0.43 | 0.53 | 0.65 | 0.6 | 0.22 | 0.14 | 0.13 |
| mamba | 0.12 | 0.17 | 0.24 | 0.34 | 0.44 | 0.54 | 0.64 | 0.69 | 0.7 | 0.64 | 0.59 |
| retnet | 0.13 | 0.17 | 0.25 | 0.34 | 0.44 | 0.53 | 0.53 | 0.45 | 0.46 | 0.43 | 0.45 |
| rwkv | 0.12 | 0.17 | 0.23 | 0.35 | 0.43 | 0.53 | 0.64 | 0.63 | 0.65 | 0.61 | 0.64 |
| gpt2 | 0.13 | 0.18 | 0.23 | 0.35 | 0.43 | 0.52 | 0.48 | 0.45 | 0.33 | 0.29 | 0.2 |
| llama2 | 0.14 | 0.17 | 0.24 | 0.34 | 0.43 | 0.53 | 0.64 | 0.62 | 0.27 | 0.14 | 0.13 |

$k = 8$, best training run

Table 11: Multiclass classification best accuracy. See Figure 1 for line plots of the same data

Query Index

| | $2^0$ | $2^1$ | $2^2$ | $2^3$ | $2^4$ | $2^5$ | $2^6$ | $2^7$ | $2^8$ | $2^9$ | $2^{10}$ |
|---|---|---|---|---|---|---|---|---|---|---|---|
| rnn | 0.5 | 0.67 | 0.78 | 0.85 | 0.86 | 0.89 | 0.9 | 0.89 | 0.88 | 0.89 | 0.89 |
| lstm | 0.5 | 0.59 | 0.65 | 0.69 | 0.69 | 0.72 | 0.71 | 0.71 | 0.71 | 0.72 | 0.71 |
| gru | 0.51 | 0.63 | 0.71 | 0.77 | 0.78 | 0.81 | 0.81 | 0.81 | 0.8 | 0.81 | 0.8 |
| lightconv | 0.51 | 0.64 | 0.73 | 0.8 | 0.82 | 0.85 | 0.84 | 0.84 | 0.83 | 0.84 | 0.84 |
| dynamicconv | 0.51 | 0.64 | 0.74 | 0.82 | 0.83 | 0.87 | 0.86 | 0.86 | 0.85 | 0.86 | 0.86 |
| s4 | 0.51 | 0.64 | 0.72 | 0.79 | 0.8 | 0.85 | 0.84 | 0.81 | 0.75 | 0.7 | 0.67 |
| h3 | 0.51 | 0.69 | 0.8 | 0.88 | 0.89 | 0.94 | 0.91 | 0.86 | 0.81 | 0.73 | 0.67 |
| hyena | 0.5 | 0.69 | 0.8 | 0.88 | 0.89 | 0.94 | 0.93 | 0.9 | 0.72 | 0.6 | 0.56 |
| mamba | 0.51 | 0.69 | 0.8 | 0.88 | 0.89 | 0.94 | 0.93 | 0.92 | 0.83 | 0.75 | 0.7 |
| retnet | 0.51 | 0.69 | 0.8 | 0.88 | 0.89 | 0.94 | 0.93 | 0.92 | 0.92 | 0.93 | 0.92 |
| rwkv | 0.51 | 0.69 | 0.8 | 0.88 | 0.89 | 0.94 | 0.93 | 0.93 | 0.93 | 0.94 | 0.93 |
| gpt2 | 0.51 | 0.69 | 0.8 | 0.88 | 0.89 | 0.94 | 0.91 | 0.88 | 0.82 | 0.75 | 0.67 |
| llama2 | 0.51 | 0.69 | 0.8 | 0.88 | 0.89 | 0.94 | 0.92 | 0.89 | 0.81 | 0.67 | 0.61 |

$k = 2$, average over all training runs

Query Index

| | $2^0$ | $2^1$ | $2^2$ | $2^3$ | $2^4$ | $2^5$ | $2^6$ | $2^7$ | $2^8$ | $2^9$ | $2^{10}$ |
|---|---|---|---|---|---|---|---|---|---|---|---|
| rnn | 0.25 | 0.33 | 0.38 | 0.4 | 0.42 | 0.44 | 0.45 | 0.44 | 0.44 | 0.45 | 0.45 |
| lstm | 0.26 | 0.3 | 0.34 | 0.38 | 0.41 | 0.44 | 0.46 | 0.45 | 0.45 | 0.46 | 0.44 |
| gru | 0.26 | 0.3 | 0.34 | 0.38 | 0.41 | 0.43 | 0.46 | 0.45 | 0.45 | 0.45 | 0.44 |
| lightconv | 0.25 | 0.31 | 0.34 | 0.36 | 0.37 | 0.39 | 0.39 | 0.38 | 0.39 | 0.4 | 0.39 |
| dynamicconv | 0.25 | 0.31 | 0.35 | 0.38 | 0.4 | 0.42 | 0.44 | 0.43 | 0.44 | 0.44 | 0.43 |
| s4 | 0.25 | 0.31 | 0.35 | 0.38 | 0.4 | 0.42 | 0.43 | 0.38 | 0.35 | 0.33 | 0.31 |
| h3 | 0.26 | 0.37 | 0.49 | 0.61 | 0.7 | 0.78 | 0.77 | 0.75 | 0.71 | 0.68 | 0.63 |
| hyena | 0.27 | 0.37 | 0.49 | 0.6 | 0.69 | 0.77 | 0.81 | 0.77 | 0.52 | 0.32 | 0.27 |
| mamba | 0.25 | 0.37 | 0.49 | 0.62 | 0.7 | 0.78 | 0.83 | 0.83 | 0.75 | 0.66 | 0.59 |
| retnet | 0.26 | 0.37 | 0.49 | 0.62 | 0.7 | 0.78 | 0.73 | 0.67 | 0.65 | 0.67 | 0.67 |
| rwkv | 0.26 | 0.37 | 0.49 | 0.62 | 0.7 | 0.78 | 0.83 | 0.83 | 0.81 | 0.83 | 0.82 |
| gpt2 | 0.26 | 0.37 | 0.49 | 0.62 | 0.7 | 0.78 | 0.73 | 0.67 | 0.57 | 0.47 | 0.36 |
| llama2 | 0.26 | 0.37 | 0.49 | 0.62 | 0.7 | 0.78 | 0.78 | 0.66 | 0.51 | 0.36 | 0.31 |

$k = 4$, average over all training runs

Query Index

| | $2^0$ | $2^1$ | $2^2$ | $2^3$ | $2^4$ | $2^5$ | $2^6$ | $2^7$ | $2^8$ | $2^9$ | $2^{10}$ |
|---|---|---|---|---|---|---|---|---|---|---|---|
| rnn | 0.12 | 0.14 | 0.15 | 0.15 | 0.16 | 0.15 | 0.16 | 0.15 | 0.16 | 0.16 | 0.16 |
| lstm | 0.12 | 0.13 | 0.15 | 0.16 | 0.17 | 0.18 | 0.18 | 0.19 | 0.18 | 0.19 | 0.18 |
| gru | 0.12 | 0.13 | 0.15 | 0.16 | 0.16 | 0.17 | 0.18 | 0.18 | 0.18 | 0.18 | 0.18 |
| lightconv | 0.13 | 0.13 | 0.14 | 0.14 | 0.14 | 0.14 | 0.14 | 0.14 | 0.13 | 0.14 | 0.14 |
| dynamicconv | 0.12 | 0.14 | 0.15 | 0.15 | 0.15 | 0.15 | 0.15 | 0.14 | 0.15 | 0.15 | 0.15 |
| s4 | 0.12 | 0.14 | 0.15 | 0.14 | 0.14 | 0.15 | 0.15 | 0.15 | 0.14 | 0.14 | 0.14 |
| h3 | 0.12 | 0.17 | 0.21 | 0.29 | 0.35 | 0.44 | 0.46 | 0.44 | 0.41 | 0.36 | 0.34 |
| hyena | 0.13 | 0.16 | 0.2 | 0.28 | 0.33 | 0.4 | 0.46 | 0.42 | 0.23 | 0.16 | 0.15 |
| mamba | 0.13 | 0.18 | 0.23 | 0.34 | 0.43 | 0.54 | 0.64 | 0.62 | 0.54 | 0.41 | 0.34 |
| retnet | 0.12 | 0.16 | 0.2 | 0.26 | 0.3 | 0.37 | 0.31 | 0.24 | 0.23 | 0.23 | 0.22 |
| rwkv | 0.13 | 0.17 | 0.23 | 0.34 | 0.43 | 0.53 | 0.62 | 0.62 | 0.63 | 0.6 | 0.61 |
| gpt2 | 0.12 | 0.16 | 0.17 | 0.19 | 0.21 | 0.24 | 0.21 | 0.19 | 0.16 | 0.15 | 0.14 |
| llama2 | 0.13 | 0.18 | 0.22 | 0.33 | 0.42 | 0.52 | 0.5 | 0.36 | 0.25 | 0.18 | 0.16 |

$k = 8$, average over all training runs

Table 12: Multiclass classification average accuracy. See Figure 5 for line plots of the same data.

Training Step — P(bursty)=0.0, best training run

| | 0 | 10000 | 20000 | 30000 | 40000 | 50000 | 60000 | 70000 | 80000 | 90000 | 100000 |
|---|---|---|---|---|---|---|---|---|---|---|---|
| gpt2 | 0.49 | 0.49 | 0.5 | 0.5 | 0.49 | 0.51 | 0.49 | 0.47 | 0.47 | 0.48 | 0.51 |
| llama2 | 0.5 | 0.52 | 0.52 | 0.52 | 0.52 | 0.47 | 0.5 | 0.5 | 0.49 | 0.48 | 0.5 |
| lightconv | 0.47 | 0.51 | 0.51 | 0.53 | 0.52 | 0.46 | 0.5 | 0.52 | 0.49 | 0.48 | 0.49 |
| dynamicconv | 0.51 | 0.49 | 0.48 | 0.51 | 0.52 | 0.51 | 0.52 | 0.52 | 0.51 | 0.47 | 0.51 |
| mn | 0.5 | 0.51 | 0.51 | 0.5 | 0.51 | 0.52 | 0.52 | 0.47 | 0.5 | 0.48 | 0.52 |
| lstm | 0.5 | 0.49 | 0.49 | 0.49 | 0.51 | 0.48 | 0.49 | 0.5 | 0.53 | 0.51 | 0.53 |
| gru | 0.49 | 0.51 | 0.49 | 0.54 | 0.51 | 0.5 | 0.52 | 0.52 | 0.5 | 0.5 | 0.5 |
| s4 | 0.51 | 0.5 | 0.49 | 0.5 | 0.51 | 0.5 | 0.49 | 0.49 | 0.5 | 0.5 | 0.48 |
| h3 | 0.5 | 0.49 | 0.49 | 0.49 | 0.48 | 0.52 | 0.5 | 0.49 | 0.49 | 0.48 | 0.48 |
| hyena | 0.51 | 0.51 | 0.47 | 0.52 | 0.52 | 0.53 | 0.52 | 0.5 | 0.5 | 0.49 | 0.48 |
| mamba | 0.49 | 0.5 | 0.49 | 0.5 | 0.5 | 0.49 | 0.49 | 0.5 | 0.51 | 0.48 | 0.48 |
| retnet | 0.52 | 0.51 | 0.52 | 0.5 | 0.5 | 0.5 | 0.5 | 0.49 | 0.5 | 0.49 | 0.55 |
| rwkv | 0.5 | 0.5 | 0.49 | 0.49 | 0.47 | 0.5 | 0.51 | 0.47 | 0.48 | 0.51 | 0.51 |

Training Step — P(bursty)=0.5, best training run

| | 0 | 10000 | 20000 | 30000 | 40000 | 50000 | 60000 | 70000 | 80000 | 90000 | 100000 |
|---|---|---|---|---|---|---|---|---|---|---|---|
| gpt2 | 0.49 | 0.54 | 0.53 | 0.5 | 0.51 | 0.5 | 0.52 | 0.49 | 0.52 | 0.5 | 0.47 |
| llama2 | 0.5 | 0.93 | 0.86 | 0.79 | 0.77 | 0.73 | 0.64 | 0.6 | 0.55 | 0.59 | 0.59 |
| lightconv | 0.5 | 0.5 | 0.5 | 0.55 | 0.51 | 0.49 | 0.5 | 0.5 | 0.51 | 0.49 | 0.5 |
| dynamicconv | 0.49 | 0.51 | 0.52 | 0.5 | 0.5 | 0.52 | 0.49 | 0.49 | 0.52 | 0.51 | 0.49 |
| mn | 0.47 | 0.5 | 0.49 | 0.49 | 0.57 | 0.52 | 0.5 | 0.5 | 0.47 | 0.46 | 0.52 |
| lstm | 0.49 | 0.56 | 0.55 | 0.54 | 0.54 | 0.57 | 0.53 | 0.49 | 0.49 | 0.49 | 0.51 |
| gru | 0.48 | 0.5 | 0.58 | 0.61 | 0.62 | 0.69 | 0.61 | 0.6 | 0.6 | 0.57 | 0.52 |
| s4 | 0.49 | 0.49 | 0.5 | 0.46 | 0.47 | 0.48 | 0.51 | 0.53 | 0.47 | 0.49 | 0.49 |
| h3 | 0.47 | 0.49 | 0.47 | 0.56 | 0.6 | 0.57 | 0.65 | 0.66 | 0.55 | 0.55 | 0.54 |
| hyena | 0.51 | 0.55 | 0.8 | 0.89 | 0.93 | 0.89 | 0.88 | 0.85 | 0.83 | 0.81 | 0.81 |
| mamba | 0.49 | 0.51 | 0.52 | 0.65 | 0.79 | 0.74 | 0.73 | 0.67 | 0.68 | 0.66 | 0.66 |
| retnet | 0.49 | 0.51 | 0.47 | 0.5 | 0.51 | 0.49 | 0.51 | 0.5 | 0.49 | 0.5 | 0.5 |
| rwkv | 0.49 | 0.56 | 0.52 | 0.57 | 0.62 | 0.62 | 0.66 | 0.65 | 0.57 | 0.57 | 0.58 |

Training Step — P(bursty)=0.9, best training run

| | 0 | 10000 | 20000 | 30000 | 40000 | 50000 | 60000 | 70000 | 80000 | 90000 | 100000 |
|---|---|---|---|---|---|---|---|---|---|---|---|
| gpt2 | 0.49 | 0.86 | 0.93 | 0.94 | 0.93 | 0.91 | 0.8 | 0.78 | 0.77 | 0.68 | 0.7 |
| llama2 | 0.51 | 0.94 | 0.93 | 0.94 | 0.95 | 0.93 | 0.95 | 0.83 | 0.91 | 0.91 | 0.9 |
| lightconv | 0.5 | 0.5 | 0.54 | 0.49 | 0.48 | 0.48 | 0.51 | 0.49 | 0.52 | 0.53 | 0.5 |
| dynamicconv | 0.51 | 0.51 | 0.55 | 0.49 | 0.54 | 0.51 | 0.5 | 0.5 | 0.52 | 0.51 | 0.53 |
| mn | 0.48 | 0.48 | 0.51 | 0.68 | 0.67 | 0.58 | 0.54 | 0.54 | 0.5 | 0.5 | 0.53 |
| lstm | 0.51 | 0.52 | 0.59 | 0.67 | 0.62 | 0.61 | 0.57 | 0.53 | 0.57 | 0.53 | 0.48 |
| gru | 0.49 | 0.5 | 0.69 | 0.82 | 0.79 | 0.73 | 0.65 | 0.6 | 0.58 | 0.6 | 0.57 |
| s4 | 0.51 | 0.5 | 0.55 | 0.56 | 0.54 | 0.55 | 0.52 | 0.52 | 0.52 | 0.54 | 0.5 |
| h3 | 0.47 | 0.84 | 0.86 | 0.91 | 0.93 | 0.93 | 0.9 | 0.91 | 0.83 | 0.83 | 0.84 |
| hyena | 0.51 | 0.81 | 0.93 | 0.95 | 0.94 | 0.93 | 0.94 | 0.9 | 0.86 | 0.8 | 0.78 |
| mamba | 0.49 | 0.5 | 0.69 | 0.92 | 0.93 | 0.94 | 0.94 | 0.91 | 0.94 | 0.93 | 0.93 |
| retnet | 0.52 | 0.52 | 0.57 | 0.74 | 0.87 | 0.88 | 0.85 | 0.82 | 0.82 | 0.78 | 0.8 |
| rwkv | 0.5 | 0.77 | 0.79 | 0.67 | 0.6 | 0.62 | 0.56 | 0.56 | 0.54 | 0.59 | 0.55 |

Training Step — P(bursty)=1.0, best training run

| | 0 | 10000 | 20000 | 30000 | 40000 | 50000 | 60000 | 70000 | 80000 | 90000 | 100000 |
|---|---|---|---|---|---|---|---|---|---|---|---|
| gpt2 | 0.5 | 0.66 | 0.93 | 0.94 | 0.93 | 0.96 | 0.91 | 0.95 | 0.95 | 0.93 | 0.96 |
| llama2 | 0.48 | 0.95 | 0.93 | 0.96 | 0.94 | 0.95 | 0.95 | 0.96 | 0.95 | 0.95 | 0.95 |
| lightconv | 0.5 | 0.52 | 0.49 | 0.5 | 0.53 | 0.49 | 0.5 | 0.48 | 0.49 | 0.48 | 0.47 |
| dynamicconv | 0.49 | 0.48 | 0.56 | 0.56 | 0.62 | 0.63 | 0.64 | 0.57 | 0.58 | 0.61 | 0.59 |
| mn | 0.48 | 0.5 | 0.66 | 0.72 | 0.69 | 0.62 | 0.59 | 0.6 | 0.6 | 0.57 | 0.58 |
| lstm | 0.51 | 0.51 | 0.57 | 0.62 | 0.66 | 0.7 | 0.71 | 0.71 | 0.71 | 0.74 | 0.73 |
| gru | 0.5 | 0.51 | 0.55 | 0.56 | 0.62 | 0.75 | 0.87 | 0.87 | 0.83 | 0.85 | 0.83 |
| s4 | 0.51 | 0.51 | 0.49 | 0.62 | 0.64 | 0.71 | 0.65 | 0.7 | 0.71 | 0.72 | 0.72 |
| h3 | 0.47 | 0.82 | 0.89 | 0.83 | 0.9 | 0.93 | 0.91 | 0.94 | 0.86 | 0.86 | 0.87 |
| hyena | 0.5 | 0.91 | 0.91 | 0.87 | 0.89 | 0.95 | 0.94 | 0.92 | 0.95 | 0.95 | 0.93 |
| mamba | 0.45 | 0.79 | 0.91 | 0.94 | 0.96 | 0.95 | 0.95 | 0.94 | 0.95 | 0.95 | 0.94 |
| retnet | 0.52 | 0.51 | 0.5 | 0.63 | 0.74 | 0.76 | 0.86 | 0.85 | 0.87 | 0.83 | 0.83 |
| rwkv | 0.53 | 0.69 | 0.83 | 0.87 | 0.9 | 0.94 | 0.94 | 0.91 | 0.84 | 0.9 | 0.87 |

Table 13: Image classification max accuracy. See Figure 8 for line plots of the same data.

Training Step — P(bursty)=0.0, average over all training runs

| | 0 | 10000 | 20000 | 30000 | 40000 | 50000 | 60000 | 70000 | 80000 | 90000 | 100000 |
|---|---|---|---|---|---|---|---|---|---|---|---|
| gpt2 | 0.5 | 0.5 | 0.5 | 0.5 | 0.49 | 0.5 | 0.5 | 0.5 | 0.5 | 0.5 | 0.49 |
| llama2 | 0.49 | 0.5 | 0.5 | 0.5 | 0.5 | 0.5 | 0.5 | 0.5 | 0.5 | 0.5 | 0.5 |
| lightconv | 0.49 | 0.49 | 0.5 | 0.5 | 0.5 | 0.5 | 0.5 | 0.5 | 0.5 | 0.5 | 0.5 |
| dynamicconv | 0.5 | 0.5 | 0.5 | 0.5 | 0.5 | 0.5 | 0.5 | 0.5 | 0.5 | 0.5 | 0.5 |
| mn | 0.5 | 0.5 | 0.5 | 0.5 | 0.5 | 0.5 | 0.5 | 0.5 | 0.5 | 0.5 | 0.5 |
| lstm | 0.5 | 0.5 | 0.5 | 0.5 | 0.5 | 0.5 | 0.49 | 0.5 | 0.5 | 0.5 | 0.5 |
| gru | 0.5 | 0.5 | 0.5 | 0.5 | 0.5 | 0.5 | 0.5 | 0.5 | 0.5 | 0.5 | 0.5 |
| s4 | 0.5 | 0.5 | 0.5 | 0.5 | 0.5 | 0.5 | 0.5 | 0.5 | 0.5 | 0.5 | 0.5 |
| h3 | 0.49 | 0.5 | 0.5 | 0.5 | 0.5 | 0.5 | 0.5 | 0.5 | 0.5 | 0.5 | 0.51 |
| hyena | 0.5 | 0.5 | 0.5 | 0.51 | 0.5 | 0.5 | 0.5 | 0.5 | 0.5 | 0.5 | 0.5 |
| mamba | 0.49 | 0.5 | 0.5 | 0.5 | 0.5 | 0.5 | 0.5 | 0.5 | 0.5 | 0.5 | 0.5 |
| retnet | 0.5 | 0.5 | 0.5 | 0.49 | 0.51 | 0.5 | 0.5 | 0.5 | 0.5 | 0.5 | 0.5 |
| rwkv | 0.51 | 0.5 | 0.5 | 0.5 | 0.49 | 0.5 | 0.5 | 0.5 | 0.5 | 0.5 | 0.5 |

Training Step — P(bursty)=0.5, average over all training runs

| | 0 | 10000 | 20000 | 30000 | 40000 | 50000 | 60000 | 70000 | 80000 | 90000 | 100000 |
|---|---|---|---|---|---|---|---|---|---|---|---|
| gpt2 | 0.5 | 0.52 | 0.51 | 0.51 | 0.5 | 0.5 | 0.5 | 0.5 | 0.5 | 0.5 | 0.5 |
| llama2 | 0.49 | 0.67 | 0.77 | 0.75 | 0.71 | 0.69 | 0.66 | 0.64 | 0.63 | 0.62 | 0.62 |
| lightconv | 0.49 | 0.5 | 0.5 | 0.5 | 0.5 | 0.5 | 0.5 | 0.5 | 0.5 | 0.5 | 0.5 |
| dynamicconv | 0.5 | 0.5 | 0.5 | 0.5 | 0.5 | 0.5 | 0.5 | 0.5 | 0.5 | 0.5 | 0.5 |
| mn | 0.5 | 0.5 | 0.5 | 0.5 | 0.5 | 0.5 | 0.5 | 0.5 | 0.5 | 0.5 | 0.5 |
| lstm | 0.5 | 0.5 | 0.5 | 0.5 | 0.51 | 0.5 | 0.5 | 0.5 | 0.5 | 0.5 | 0.5 |
| gru | 0.5 | 0.5 | 0.51 | 0.51 | 0.51 | 0.51 | 0.51 | 0.52 | 0.52 | 0.52 | 0.52 |
| s4 | 0.5 | 0.5 | 0.5 | 0.5 | 0.5 | 0.49 | 0.5 | 0.5 | 0.49 | 0.5 | 0.5 |
| h3 | 0.49 | 0.5 | 0.5 | 0.5 | 0.51 | 0.51 | 0.52 | 0.51 | 0.51 | 0.51 | 0.51 |
| hyena | 0.5 | 0.56 | 0.66 | 0.7 | 0.7 | 0.69 | 0.68 | 0.66 | 0.65 | 0.65 | 0.65 |
| mamba | 0.49 | 0.51 | 0.51 | 0.54 | 0.55 | 0.56 | 0.55 | 0.55 | 0.54 | 0.53 | 0.54 |
| retnet | 0.5 | 0.5 | 0.5 | 0.5 | 0.51 | 0.5 | 0.5 | 0.5 | 0.5 | 0.5 | 0.49 |
| rwkv | 0.51 | 0.53 | 0.54 | 0.54 | 0.53 | 0.53 | 0.53 | 0.53 | 0.52 | 0.51 | 0.51 |

Training Step — P(bursty)=0.9, average over all training runs

| | 0 | 10000 | 20000 | 30000 | 40000 | 50000 | 60000 | 70000 | 80000 | 90000 | 100000 |
|---|---|---|---|---|---|---|---|---|---|---|---|
| gpt2 | 0.5 | 0.59 | 0.68 | 0.72 | 0.71 | 0.7 | 0.69 | 0.67 | 0.65 | 0.63 | 0.63 |
| llama2 | 0.49 | 0.77 | 0.9 | 0.91 | 0.91 | 0.91 | 0.9 | 0.9 | 0.89 | 0.88 | 0.87 |
| lightconv | 0.49 | 0.5 | 0.5 | 0.5 | 0.5 | 0.5 | 0.5 | 0.5 | 0.5 | 0.5 | 0.5 |
| dynamicconv | 0.5 | 0.5 | 0.5 | 0.5 | 0.5 | 0.5 | 0.51 | 0.5 | 0.5 | 0.5 | 0.49 |
| mn | 0.5 | 0.5 | 0.51 | 0.52 | 0.52 | 0.52 | 0.52 | 0.52 | 0.52 | 0.52 | 0.51 |
| lstm | 0.5 | 0.51 | 0.51 | 0.51 | 0.52 | 0.51 | 0.51 | 0.51 | 0.51 | 0.52 | 0.51 |
| gru | 0.5 | 0.5 | 0.53 | 0.56 | 0.56 | 0.56 | 0.56 | 0.56 | 0.56 | 0.55 | 0.54 |
| s4 | 0.5 | 0.5 | 0.5 | 0.51 | 0.5 | 0.51 | 0.5 | 0.5 | 0.5 | 0.51 | 0.5 |
| h3 | 0.49 | 0.55 | 0.64 | 0.69 | 0.69 | 0.69 | 0.68 | 0.68 | 0.66 | 0.65 | 0.64 |
| hyena | 0.5 | 0.69 | 0.85 | 0.89 | 0.9 | 0.89 | 0.88 | 0.85 | 0.85 | 0.84 | 0.83 |
| mamba | 0.49 | 0.52 | 0.58 | 0.66 | 0.69 | 0.7 | 0.72 | 0.72 | 0.72 | 0.72 | 0.72 |
| retnet | 0.5 | 0.5 | 0.51 | 0.51 | 0.52 | 0.52 | 0.51 | 0.52 | 0.51 | 0.51 | 0.51 |
| rwkv | 0.51 | 0.64 | 0.69 | 0.66 | 0.65 | 0.62 | 0.6 | 0.59 | 0.57 | 0.56 | 0.56 |

Training Step — P(bursty)=1.0, average over all training runs

| | 0 | 10000 | 20000 | 30000 | 40000 | 50000 | 60000 | 70000 | 80000 | 90000 | 100000 |
|---|---|---|---|---|---|---|---|---|---|---|---|
| gpt2 | 0.5 | 0.62 | 0.75 | 0.81 | 0.86 | 0.88 | 0.88 | 0.89 | 0.89 | 0.9 | 0.89 |
| llama2 | 0.49 | 0.78 | 0.91 | 0.93 | 0.94 | 0.94 | 0.94 | 0.94 | 0.94 | 0.94 | 0.94 |
| lightconv | 0.49 | 0.5 | 0.5 | 0.5 | 0.5 | 0.5 | 0.5 | 0.49 | 0.5 | 0.5 | 0.5 |
| dynamicconv | 0.5 | 0.5 | 0.51 | 0.51 | 0.5 | 0.51 | 0.51 | 0.5 | 0.5 | 0.5 | 0.5 |
| mn | 0.5 | 0.52 | 0.55 | 0.56 | 0.54 | 0.55 | 0.54 | 0.54 | 0.53 | 0.54 | 0.53 |
| lstm | 0.5 | 0.5 | 0.52 | 0.53 | 0.54 | 0.55 | 0.54 | 0.54 | 0.54 | 0.54 | 0.54 |
| gru | 0.5 | 0.51 | 0.54 | 0.56 | 0.56 | 0.58 | 0.58 | 0.59 | 0.58 | 0.58 | 0.58 |
| s4 | 0.5 | 0.5 | 0.51 | 0.52 | 0.52 | 0.53 | 0.53 | 0.53 | 0.52 | 0.53 | 0.53 |
| h3 | 0.49 | 0.6 | 0.67 | 0.73 | 0.74 | 0.75 | 0.75 | 0.77 | 0.77 | 0.77 | 0.77 |
| hyena | 0.5 | 0.73 | 0.83 | 0.86 | 0.87 | 0.88 | 0.88 | 0.89 | 0.89 | 0.89 | 0.9 |
| mamba | 0.49 | 0.54 | 0.62 | 0.68 | 0.7 | 0.73 | 0.73 | 0.73 | 0.73 | 0.73 | 0.73 |
| retnet | 0.5 | 0.5 | 0.5 | 0.5 | 0.51 | 0.51 | 0.51 | 0.51 | 0.52 | 0.52 | 0.51 |
| rwkv | 0.51 | 0.66 | 0.76 | 0.77 | 0.79 | 0.79 | 0.79 | 0.78 | 0.78 | 0.78 | 0.77 |

Table 14: Image classification average accuracy. See Figure 2 for line plots of the same data.

Query Index

| | $2^0$ | $2^1$ | $2^2$ | $2^3$ | $2^4$ | $2^5$ | $2^6$ | $2^7$ | $2^8$ | $2^9$ | $2^{10}$ |
|---|---|---|---|---|---|---|---|---|---|---|---|
| tran-no-pos-emb | 0.07 | 0.17 | 0.33 | 0.58 | 0.87 | 0.99 | 1.0 | 1.0 | 0.98 | 0.83 | 0.54 |
| tran-sinu | 0.06 | 0.15 | 0.32 | 0.57 | 0.86 | 0.99 | 1.0 | 1.0 | 0.98 | 0.89 | 0.73 |
| tran-abs-pos-emb | 0.05 | 0.16 | 0.31 | 0.58 | 0.86 | 0.99 | 0.97 | 0.79 | 0.62 | 0.55 | 0.4 |
| tran-rope | 0.05 | 0.15 | 0.31 | 0.58 | 0.87 | 0.99 | 0.97 | 0.7 | 0.28 | 0.15 | 0.08 |
| tran-alibi | 0.05 | 0.16 | 0.31 | 0.58 | 0.86 | 1.0 | 0.99 | 0.88 | 0.77 | 0.32 | 0.22 |

$|V| = 20$, best training run

Query Index

| | $2^0$ | $2^1$ | $2^2$ | $2^3$ | $2^4$ | $2^5$ | $2^6$ | $2^7$ | $2^8$ | $2^9$ | $2^{10}$ |
|---|---|---|---|---|---|---|---|---|---|---|---|
| tran-no-pos-emb | 0.04 | 0.1 | 0.25 | 0.4 | 0.67 | 0.92 | 1.0 | 1.0 | 0.98 | 0.93 | 0.81 |
| tran-sinu | 0.04 | 0.09 | 0.23 | 0.41 | 0.67 | 0.91 | 0.97 | 0.94 | 0.89 | 0.71 | 0.47 |
| tran-abs-pos-emb | 0.04 | 0.09 | 0.22 | 0.41 | 0.66 | 0.91 | 0.93 | 0.91 | 0.89 | 0.87 | 0.89 |
| tran-rope | 0.03 | 0.1 | 0.23 | 0.41 | 0.66 | 0.92 | 0.88 | 0.17 | 0.07 | 0.04 | 0.04 |
| tran-alibi | 0.04 | 0.09 | 0.23 | 0.41 | 0.67 | 0.91 | 0.93 | 0.91 | 0.84 | 0.51 | 0.05 |

$|V| = 30$, best training run

Query Index

| | $2^0$ | $2^1$ | $2^2$ | $2^3$ | $2^4$ | $2^5$ | $2^6$ | $2^7$ | $2^8$ | $2^9$ | $2^{10}$ |
|---|---|---|---|---|---|---|---|---|---|---|---|
| tran-no-pos-emb | 0.03 | 0.07 | 0.17 | 0.32 | 0.55 | 0.81 | 0.99 | 0.99 | 0.99 | 0.99 | 0.98 |
| tran-sinu | 0.02 | 0.07 | 0.17 | 0.32 | 0.56 | 0.81 | 0.99 | 1.0 | 1.0 | 1.0 | 1.0 |
| tran-abs-pos-emb | 0.03 | 0.08 | 0.16 | 0.32 | 0.55 | 0.8 | 0.8 | 0.22 | 0.1 | 0.06 | 0.05 |
| tran-rope | 0.03 | 0.07 | 0.17 | 0.33 | 0.55 | 0.79 | 0.98 | 0.92 | 0.73 | 0.33 | 0.16 |
| tran-alibi | 0.03 | 0.07 | 0.18 | 0.32 | 0.55 | 0.79 | 0.64 | 0.38 | 0.13 | 0.09 | 0.05 |

$|V| = 40$, best training run

Query Index

| | $2^0$ | $2^1$ | $2^2$ | $2^3$ | $2^4$ | $2^5$ | $2^6$ | $2^7$ | $2^8$ | $2^9$ | $2^{10}$ |
|---|---|---|---|---|---|---|---|---|---|---|---|
| tran-no-pos-emb | 0.06 | 0.15 | 0.31 | 0.58 | 0.87 | 0.99 | 1.0 | 1.0 | 0.99 | 0.95 | 0.88 |
| tran-sinu | 0.05 | 0.16 | 0.31 | 0.58 | 0.87 | 0.99 | 1.0 | 1.0 | 0.99 | 0.93 | 0.85 |
| tran-abs-pos-emb | 0.05 | 0.15 | 0.31 | 0.58 | 0.87 | 0.99 | 0.96 | 0.94 | 0.9 | 0.86 | 0.78 |
| tran-rope | 0.06 | 0.15 | 0.31 | 0.58 | 0.87 | 0.99 | 0.97 | 0.79 | 0.48 | 0.23 | 0.14 |
| tran-alibi | 0.05 | 0.15 | 0.31 | 0.58 | 0.87 | 0.99 | 0.96 | 0.93 | 0.87 | 0.65 | 0.35 |

$|V| = 20$, average over all training runs

Query Index

| | $2^0$ | $2^1$ | $2^2$ | $2^3$ | $2^4$ | $2^5$ | $2^6$ | $2^7$ | $2^8$ | $2^9$ | $2^{10}$ |
|---|---|---|---|---|---|---|---|---|---|---|---|
| tran-no-pos-emb | 0.04 | 0.09 | 0.23 | 0.41 | 0.66 | 0.91 | 1.0 | 0.99 | 0.98 | 0.95 | 0.89 |
| tran-sinu | 0.03 | 0.09 | 0.23 | 0.41 | 0.66 | 0.91 | 0.99 | 0.97 | 0.94 | 0.89 | 0.8 |
| tran-abs-pos-emb | 0.04 | 0.09 | 0.23 | 0.42 | 0.67 | 0.91 | 0.85 | 0.79 | 0.71 | 0.59 | 0.46 |
| tran-rope | 0.04 | 0.1 | 0.23 | 0.41 | 0.66 | 0.91 | 0.92 | 0.55 | 0.27 | 0.09 | 0.06 |
| tran-alibi | 0.03 | 0.09 | 0.23 | 0.41 | 0.67 | 0.91 | 0.84 | 0.78 | 0.64 | 0.4 | 0.24 |

$|V| = 30$, average over all training runs

Query Index

| | $2^0$ | $2^1$ | $2^2$ | $2^3$ | $2^4$ | $2^5$ | $2^6$ | $2^7$ | $2^8$ | $2^9$ | $2^{10}$ |
|---|---|---|---|---|---|---|---|---|---|---|---|
| tran-no-pos-emb | 0.03 | 0.07 | 0.16 | 0.32 | 0.55 | 0.79 | 0.96 | 0.95 | 0.91 | 0.83 | 0.72 |
| tran-sinu | 0.02 | 0.07 | 0.16 | 0.32 | 0.55 | 0.79 | 0.89 | 0.79 | 0.67 | 0.54 | 0.43 |
| tran-abs-pos-emb | 0.03 | 0.07 | 0.16 | 0.32 | 0.56 | 0.8 | 0.67 | 0.59 | 0.44 | 0.34 | 0.29 |
| tran-rope | 0.03 | 0.07 | 0.16 | 0.32 | 0.56 | 0.8 | 0.88 | 0.58 | 0.35 | 0.12 | 0.08 |
| tran-alibi | 0.03 | 0.07 | 0.17 | 0.32 | 0.56 | 0.8 | 0.67 | 0.58 | 0.37 | 0.26 | 0.16 |

$|V| = 40$, average over all training runs

Table 15: Associative recall experiments repeated across various transformer positional embedding options.

Query Index

| | $2^0$ | $2^1$ | $2^2$ | $2^3$ | $2^4$ | $2^5$ | $2^6$ | $2^7$ | $2^8$ | $2^9$ | $2^{10}$ |
|---|---|---|---|---|---|---|---|---|---|---|---|
| tran-no-pos-emb | 4.93 | 3.84 | 1.93 | 0.0 | 0.0 | 0.0 | 0.0 | 0.02 | 0.07 | 0.36 | 0.94 |
| tran-sinu | 4.93 | 3.84 | 1.93 | 0.0 | 0.0 | 0.0 | 0.85 | 0.46 | 0.75 | 1.93 | 4.19 |
| tran-abs-pos-emb | 4.93 | 3.84 | 1.92 | 0.0 | 0.0 | 0.0 | 0.12 | 0.72 | 1.9 | 3.78 | 4.47 |
| tran-rope | 4.93 | 3.84 | 1.92 | 0.0 | 0.0 | 0.0 | 0.11 | 0.34 | 0.26 | 2.17 | 3.55 |
| tran-alibi | 4.93 | 3.84 | 1.92 | 0.0 | 0.0 | 0.0 | 0.14 | 0.63 | 2.14 | 4.23 | 4.26 |

$d = 5$, best training run

Query Index

| | $2^0$ | $2^1$ | $2^2$ | $2^3$ | $2^4$ | $2^5$ | $2^6$ | $2^7$ | $2^8$ | $2^9$ | $2^{10}$ |
|---|---|---|---|---|---|---|---|---|---|---|---|
| tran-no-pos-emb | 9.69 | 8.97 | 7.23 | 2.88 | 0.0 | 0.0 | 0.03 | 0.15 | 0.42 | 0.83 | 1.56 |
| tran-sinu | 9.69 | 8.97 | 7.21 | 2.88 | 0.0 | 0.0 | 0.91 | 6.19 | 24.76 | 52.68 | 63.67 |
| tran-abs-pos-emb | 9.69 | 8.97 | 7.21 | 2.89 | 0.0 | 0.0 | 0.69 | 2.91 | 13.37 | 25.33 | 37.32 |
| tran-rope | 9.69 | 8.96 | 7.22 | 2.9 | 0.0 | 0.0 | 0.13 | 2.02 | 4.37 | 6.16 | 7.64 |
| tran-alibi | 9.69 | 8.97 | 7.21 | 2.89 | 0.01 | 0.0 | 0.5 | 1.29 | 9.14 | 36.19 | 48.8 |

$d = 10$, best training run

Query Index

| | $2^0$ | $2^1$ | $2^2$ | $2^3$ | $2^4$ | $2^5$ | $2^6$ | $2^7$ | $2^8$ | $2^9$ | $2^{10}$ |
|---|---|---|---|---|---|---|---|---|---|---|---|
| tran-no-pos-emb | 19.52 | 19.28 | 17.3 | 12.96 | 5.41 | 0.01 | 0.34 | 3.06 | 7.9 | 12.57 | 16.47 |
| tran-sinu | 19.52 | 19.3 | 17.3 | 12.92 | 5.36 | 0.01 | 12.99 | 51.44 | 95.08 | 142.31 | 153.12 |
| tran-abs-pos-emb | 19.52 | 19.29 | 17.31 | 12.96 | 5.41 | 0.01 | 107.49 | 172.36 | 187.04 | 218.81 | 246.92 |
| tran-rope | 19.52 | 19.29 | 17.28 | 12.98 | 5.38 | 0.01 | 4.97 | 17.0 | 35.85 | 38.77 | 34.39 |
| tran-alibi | 19.52 | 19.3 | 17.28 | 12.96 | 5.37 | 0.01 | 46.68 | 31.05 | 28.29 | 25.33 | 28.69 |

$d = 20$, best training run

Query Index

| | $2^0$ | $2^1$ | $2^2$ | $2^3$ | $2^4$ | $2^5$ | $2^6$ | $2^7$ | $2^8$ | $2^9$ | $2^{10}$ |
|---|---|---|---|---|---|---|---|---|---|---|---|
| tran-no-pos-emb | 4.93 | 3.85 | 1.95 | 0.09 | 0.02 | 0.01 | 0.02 | 0.08 | 0.26 | 0.62 | 1.05 |
| tran-sinu | 4.93 | 3.85 | 1.93 | 0.06 | 0.0 | 0.0 | 0.52 | 0.93 | 1.49 | 2.75 | 3.93 |
| tran-abs-pos-emb | 4.93 | 3.84 | 1.93 | 0.04 | 0.0 | 0.0 | 1.36 | 2.37 | 2.73 | 3.76 | 5.14 |
| tran-rope | 4.93 | 3.84 | 1.93 | 0.05 | 0.0 | 0.0 | 0.21 | 0.94 | 1.62 | 3.08 | 4.26 |
| tran-alibi | 4.93 | 3.85 | 1.93 | 0.04 | 0.0 | 0.0 | 1.22 | 1.8 | 2.83 | 4.41 | 5.87 |

$d = 5$, average over all training runs

Query Index

| | $2^0$ | $2^1$ | $2^2$ | $2^3$ | $2^4$ | $2^5$ | $2^6$ | $2^7$ | $2^8$ | $2^9$ | $2^{10}$ |
|---|---|---|---|---|---|---|---|---|---|---|---|
| tran-no-pos-emb | 9.69 | 8.97 | 7.27 | 3.06 | 0.25 | 0.09 | 0.17 | 0.69 | 2.14 | 4.78 | 6.69 |
| tran-sinu | 9.69 | 8.96 | 7.23 | 2.97 | 0.12 | 0.02 | 2.51 | 6.08 | 11.54 | 19.18 | 24.37 |
| tran-abs-pos-emb | 9.69 | 8.96 | 7.23 | 2.96 | 0.1 | 0.02 | 8.04 | 9.98 | 11.0 | 14.4 | 15.53 |
| tran-rope | 9.69 | 8.96 | 7.24 | 2.99 | 0.13 | 0.02 | 1.34 | 5.95 | 10.83 | 16.69 | 20.08 |
| tran-alibi | 9.69 | 8.96 | 7.24 | 2.96 | 0.1 | 0.02 | 7.4 | 9.83 | 10.8 | 15.01 | 18.03 |

$d = 10$, average over all training runs

Query Index

| | $2^0$ | $2^1$ | $2^2$ | $2^3$ | $2^4$ | $2^5$ | $2^6$ | $2^7$ | $2^8$ | $2^9$ | $2^{10}$ |
|---|---|---|---|---|---|---|---|---|---|---|---|
| tran-no-pos-emb | 19.53 | 19.36 | 17.34 | 13.17 | 6.16 | 1.09 | 4.7 | 18.84 | 23.09 | 28.07 | 30.83 |
| tran-sinu | 19.52 | 19.35 | 17.31 | 13.01 | 5.78 | 0.44 | 28.85 | 52.47 | 73.79 | 83.03 | 91.84 |
| tran-abs-pos-emb | 19.52 | 19.31 | 17.3 | 13.01 | 5.81 | 0.5 | 69.64 | 83.79 | 80.78 | 87.4 | 86.99 |
| tran-rope | 19.52 | 19.34 | 17.3 | 12.99 | 5.7 | 0.37 | 16.06 | 40.23 | 43.75 | 63.07 | 74.12 |
| tran-alibi | 19.52 | 19.31 | 17.31 | 13.0 | 5.8 | 0.48 | 67.85 | 87.54 | 89.73 | 91.89 | 100.37 |

$d = 20$, average over all training runs

Table 16: Linear regression experiments repeated across various transformer positional embedding options.

### Query Index

|  | $2^0$ | $2^1$ | $2^2$ | $2^3$ | $2^4$ | $2^5$ | $2^6$ | $2^7$ | $2^8$ | $2^9$ | $2^{10}$ |
|---|---|---|---|---|---|---|---|---|---|---|---|
| tran-no-pos-emb | 0.52 | 0.7 | 0.8 | 0.88 | 0.89 | 0.94 | 0.92 | 0.94 | 0.92 | 0.94 | 0.92 |
| tran-sinu | 0.5 | 0.69 | 0.8 | 0.88 | 0.89 | 0.94 | 0.91 | 0.91 | 0.9 | 0.91 | 0.87 |
| tran-abs-pos-emb | 0.51 | 0.69 | 0.8 | 0.87 | 0.89 | 0.94 | 0.92 | 0.89 | 0.81 | 0.71 | 0.59 |
| tran-rope | 0.51 | 0.7 | 0.8 | 0.87 | 0.89 | 0.94 | 0.93 | 0.93 | 0.91 | 0.87 | 0.7 |
| tran-alibi | 0.51 | 0.69 | 0.8 | 0.87 | 0.89 | 0.94 | 0.92 | 0.91 | 0.83 | 0.67 | 0.58 |

$k = 2$, best training run

### Query Index

|  | $2^0$ | $2^1$ | $2^2$ | $2^3$ | $2^4$ | $2^5$ | $2^6$ | $2^7$ | $2^8$ | $2^9$ | $2^{10}$ |
|---|---|---|---|---|---|---|---|---|---|---|---|
| tran-no-pos-emb | 0.25 | 0.38 | 0.47 | 0.63 | 0.71 | 0.78 | 0.83 | 0.84 | 0.83 | 0.82 | 0.81 |
| tran-sinu | 0.25 | 0.38 | 0.48 | 0.62 | 0.71 | 0.78 | 0.82 | 0.81 | 0.8 | 0.77 | 0.68 |
| tran-abs-pos-emb | 0.27 | 0.37 | 0.49 | 0.62 | 0.7 | 0.78 | 0.77 | 0.75 | 0.7 | 0.7 | 0.52 |
| tran-rope | 0.27 | 0.38 | 0.51 | 0.63 | 0.7 | 0.78 | 0.84 | 0.84 | 0.8 | 0.82 | 0.77 |
| tran-alibi | 0.27 | 0.36 | 0.49 | 0.63 | 0.7 | 0.79 | 0.76 | 0.73 | 0.71 | 0.68 | 0.54 |

$k = 4$, best training run

### Query Index

|  | $2^0$ | $2^1$ | $2^2$ | $2^3$ | $2^4$ | $2^5$ | $2^6$ | $2^7$ | $2^8$ | $2^9$ | $2^{10}$ |
|---|---|---|---|---|---|---|---|---|---|---|---|
| tran-no-pos-emb | 0.12 | 0.18 | 0.22 | 0.34 | 0.44 | 0.53 | 0.62 | 0.65 | 0.56 | 0.43 | 0.26 |
| tran-sinu | 0.12 | 0.18 | 0.23 | 0.34 | 0.44 | 0.54 | 0.63 | 0.66 | 0.59 | 0.45 | 0.3 |
| tran-abs-pos-emb | 0.09 | 0.17 | 0.24 | 0.34 | 0.44 | 0.54 | 0.54 | 0.53 | 0.49 | 0.42 | 0.4 |
| tran-rope | 0.12 | 0.18 | 0.23 | 0.35 | 0.44 | 0.54 | 0.59 | 0.57 | 0.43 | 0.3 | 0.24 |
| tran-alibi | 0.13 | 0.18 | 0.22 | 0.34 | 0.44 | 0.53 | 0.55 | 0.51 | 0.41 | 0.25 | 0.21 |

$k = 8$, best training run

### Query Index

|  | $2^0$ | $2^1$ | $2^2$ | $2^3$ | $2^4$ | $2^5$ | $2^6$ | $2^7$ | $2^8$ | $2^9$ | $2^{10}$ |
|---|---|---|---|---|---|---|---|---|---|---|---|
| tran-no-pos-emb | 0.51 | 0.69 | 0.8 | 0.88 | 0.89 | 0.94 | 0.92 | 0.93 | 0.91 | 0.92 | 0.88 |
| tran-sinu | 0.51 | 0.7 | 0.8 | 0.88 | 0.89 | 0.94 | 0.92 | 0.91 | 0.87 | 0.84 | 0.76 |
| tran-abs-pos-emb | 0.51 | 0.69 | 0.8 | 0.88 | 0.89 | 0.94 | 0.92 | 0.91 | 0.87 | 0.85 | 0.79 |
| tran-rope | 0.51 | 0.69 | 0.8 | 0.88 | 0.89 | 0.94 | 0.93 | 0.92 | 0.89 | 0.87 | 0.78 |
| tran-alibi | 0.51 | 0.69 | 0.8 | 0.88 | 0.89 | 0.94 | 0.92 | 0.91 | 0.85 | 0.77 | 0.67 |

$k = 2$, average over all training runs

### Query Index

|  | $2^0$ | $2^1$ | $2^2$ | $2^3$ | $2^4$ | $2^5$ | $2^6$ | $2^7$ | $2^8$ | $2^9$ | $2^{10}$ |
|---|---|---|---|---|---|---|---|---|---|---|---|
| tran-no-pos-emb | 0.26 | 0.37 | 0.49 | 0.62 | 0.7 | 0.78 | 0.83 | 0.81 | 0.77 | 0.71 | 0.6 |
| tran-sinu | 0.26 | 0.37 | 0.49 | 0.62 | 0.7 | 0.78 | 0.8 | 0.77 | 0.7 | 0.6 | 0.47 |
| tran-abs-pos-emb | 0.26 | 0.37 | 0.49 | 0.62 | 0.7 | 0.78 | 0.77 | 0.74 | 0.69 | 0.62 | 0.5 |
| tran-rope | 0.26 | 0.37 | 0.49 | 0.62 | 0.7 | 0.78 | 0.82 | 0.78 | 0.71 | 0.61 | 0.51 |
| tran-alibi | 0.26 | 0.37 | 0.49 | 0.62 | 0.7 | 0.78 | 0.77 | 0.73 | 0.61 | 0.46 | 0.35 |

$k = 4$, average over all training runs

### Query Index

|  | $2^0$ | $2^1$ | $2^2$ | $2^3$ | $2^4$ | $2^5$ | $2^6$ | $2^7$ | $2^8$ | $2^9$ | $2^{10}$ |
|---|---|---|---|---|---|---|---|---|---|---|---|
| tran-no-pos-emb | 0.13 | 0.18 | 0.23 | 0.34 | 0.44 | 0.53 | 0.64 | 0.65 | 0.6 | 0.5 | 0.4 |
| tran-sinu | 0.13 | 0.18 | 0.23 | 0.34 | 0.44 | 0.54 | 0.59 | 0.56 | 0.48 | 0.36 | 0.29 |
| tran-abs-pos-emb | 0.13 | 0.18 | 0.23 | 0.34 | 0.43 | 0.51 | 0.5 | 0.48 | 0.41 | 0.33 | 0.27 |
| tran-rope | 0.13 | 0.18 | 0.23 | 0.34 | 0.43 | 0.54 | 0.58 | 0.51 | 0.37 | 0.26 | 0.2 |
| tran-alibi | 0.13 | 0.18 | 0.23 | 0.34 | 0.43 | 0.51 | 0.49 | 0.46 | 0.37 | 0.26 | 0.2 |

$k = 8$, average over all training runs

Table 17: Multiclass classification experiments repeated across various transformer positional embedding options.

| token scheme | use pos embed | model | mean squared error | | | |
|---|---|---|---|---|---|---|
| | | d= | 5 | 10 | 20 | 30 |
| concat | False | decoder | 0.000 | 0.000 | 0.003 | 3.166 |
| | | encoder | 0.000 | 0.000 | 0.000 | 0.138 |
| | True | decoder | 0.000 | 0.000 | 0.002 | 5.038 |
| | | encoder | 0.000 | 0.000 | 0.000 | 0.165 |
| sum | False | decoder | 0.752 | 1.096 | 2.818 | 6.479 |
| | | encoder | 0.724 | 1.087 | 2.281 | 5.537 |
| | True | decoder | 0.741 | 1.139 | 2.618 | 6.142 |
| | | encoder | 0.720 | 1.100 | 2.388 | 5.605 |

(a) Linear regression (best run)

| token scheme | use pos embed | model | mean squared error | | | |
|---|---|---|---|---|---|---|
| | | d= | 5 | 10 | 20 | 30 |
| concat | False | decoder | 0.000 | 0.003 | 0.196 | 6.373 |
| | | encoder | 0.000 | 0.000 | 0.008 | 3.386 |
| | True | decoder | 0.000 | 0.002 | 0.105 | 6.967 |
| | | encoder | 0.000 | 0.000 | 0.013 | 3.710 |
| sum | False | decoder | 1.270 | 1.329 | 3.356 | 7.193 |
| | | encoder | 1.243 | 1.282 | 2.736 | 6.147 |
| | True | decoder | 1.251 | 1.323 | 3.212 | 6.995 |
| | | encoder | 1.248 | 1.290 | 2.772 | 6.196 |

(b) Linear regression (average)

| token scheme | use pos embed | model | accuracy | | |
|---|---|---|---|---|---|
| | | |V|= | 20 | 30 | 40 |
| concat | False | decoder | 0.999 | 0.929 | 0.822 |
| | | encoder | 0.999 | 0.933 | 0.820 |
| | True | decoder | 0.998 | 0.927 | 0.817 |
| | | encoder | 0.999 | 0.931 | 0.820 |
| sum | False | decoder | 0.999 | 0.928 | 0.824 |
| | | encoder | 0.999 | 0.934 | 0.820 |
| | True | decoder | 0.999 | 0.934 | 0.824 |
| | | encoder | 0.999 | 0.935 | 0.817 |

(c) Associative recall (best run)

| token scheme | use pos embed | model | accuracy | | |
|---|---|---|---|---|---|
| | | |V|= | 20 | 30 | 40 |
| concat | False | decoder | 0.998 | 0.924 | 0.812 |
| | | encoder | 0.998 | 0.928 | 0.812 |
| | True | decoder | 0.997 | 0.922 | 0.811 |
| | | encoder | 0.997 | 0.925 | 0.815 |
| sum | False | decoder | 0.998 | 0.924 | 0.813 |
| | | encoder | 0.998 | 0.927 | 0.813 |
| | True | decoder | 0.998 | 0.926 | 0.815 |
| | | encoder | 0.998 | 0.929 | 0.811 |

(d) Associative recall (average)

| token scheme | use pos embed | model | accuracy | | |
|---|---|---|---|---|---|
| | | k= | 2 | 4 | 8 |
| concat | False | decoder | 0.931 | 0.784 | 0.553 |
| | | encoder | 0.934 | 0.780 | 0.556 |
| | True | decoder | 0.930 | 0.785 | 0.552 |
| | | encoder | 0.934 | 0.780 | 0.560 |
| sum | False | decoder | 0.928 | 0.782 | 0.549 |
| | | encoder | 0.931 | 0.778 | 0.557 |
| | True | decoder | 0.932 | 0.781 | 0.553 |
| | | encoder | 0.932 | 0.778 | 0.558 |

(e) Multiclass classification (best run)

| token scheme | use pos embed | model | accuracy | | |
|---|---|---|---|---|---|
| | | k= | 2 | 4 | 8 |
| concat | False | decoder | 0.929 | 0.779 | 0.549 |
| | | encoder | 0.931 | 0.777 | 0.551 |
| | True | decoder | 0.927 | 0.777 | 0.547 |
| | | encoder | 0.931 | 0.776 | 0.551 |
| sum | False | decoder | 0.926 | 0.772 | 0.541 |
| | | encoder | 0.929 | 0.770 | 0.545 |
| | True | decoder | 0.925 | 0.772 | 0.538 |
| | | encoder | 0.930 | 0.771 | 0.544 |

(f) Multiclass classification (average)

| token scheme | use pos embed | model | accuracy | |
|---|---|---|---|---|
| | | P(bursty)= | 0.9 | 1.0 |
| concat | False | decoder | 0.945 | 0.951 |
| | | encoder | 0.950 | 0.945 |
| | True | decoder | 0.942 | 0.950 |
| | | encoder | 0.938 | 0.950 |
| sum | False | decoder | 0.820 | 0.952 |
| | | encoder | 0.845 | 0.946 |
| | True | decoder | 0.792 | 0.963 |
| | | encoder | 0.756 | 0.957 |

(g) Image classification (best run)

| token scheme | use pos embed | model | accuracy | |
|---|---|---|---|---|
| | | P(bursty)= | 0.9 | 1.0 |
| concat | False | decoder | 0.842 | 0.928 |
| | | encoder | 0.853 | 0.928 |
| | True | decoder | 0.914 | 0.933 |
| | | encoder | 0.915 | 0.931 |
| sum | False | decoder | 0.680 | 0.879 |
| | | encoder | 0.666 | 0.855 |
| | True | decoder | 0.678 | 0.927 |
| | | encoder | 0.670 | 0.886 |

(h) Image classification (average)

Figure 11: Permutation invariance experiments.

