# OpenReview forum: "Is attention required for ICL? Exploring the Relationship Between Model Architecture and In-Context Learning Ability"
_ICLR.cc/2024/Conference — ICLR 2024 poster_

### Official Review · Reviewer_9CgZ · 2023-10-26

**Soundness:** 3 good
**Presentation:** 3 good
**Contribution:** 2 fair
**Rating:** 5
**Confidence:** 4

**Summary:**

The paper studies the existence of the ability of performing In-Context Learning (ICL) among different architectural choices like Transformers, State Space Models, RNNs and CNNs across different kinds of tasks. Typically in the regime of large language models, primarily transformer based models have been scaled up and shown to shine in ICL, but the authors show that other architectures are also capable of performing ICL, and some even better than transformers in the settings considered. To fairly test for ICL, authors train all the models from scratch, following the setup of Garg et. al (2023) where training is done on a suite of different supervised learning setups. The authors provide a clear answer that most of the architectures are capable of performing ICL and this phenomena is not specific to transformer based models, and also provide an interesting revelation of state-space models performing better ICL capabilities, which would be quite useful as such models do not incur the same computational costs as transformers.

**Strengths:**

- The work explores diverse choices over tasks as well as architecture setups, and successfully answers the question posed which is to study if each of the architectures possess some ICL capability.

- The findings highlight an interesting result which is the superior performance of state-space based models like HYENA when compared to attention-based models, which provides clear validation of the need to scale such state-space models on real-world data like language.

- The release of code for this work would be very useful for the community to further study, analyze and understand ICL capabilities on different tasks, and when should one approach be considered over the other.

**Weaknesses:**

While the authors have done a fantastic job on studying different tasks and model architectures, I found some of the core components lacking. For example, details about how the context input is encoded and provided to the models is missing, as well as an explanation and hypothesis of why certain transformer based models do better than the others, and what the differences are. I have listed some of my specific concerns below, and would be happy to raise my score if they are addressed.

- Since the macro perspective is only studied in the language modeling domain, it would make more sense to move the write-up around it under the Language Modeling section.
- It would be nice if the authors could provide a write-up on the differences between the various transformer-based and state-space based models for completeness. In particular, why does one model based on transformers (eg. LLAMA2) give better ICL performance than the other (eg. T5).
- How does this contrast to the already present information that Transformers can do ICL considerably well? The experiments in Table 2 seem to show that attention-based models are fairly poor at ICL but Garg et. al (2023) and Muller et. al (2022) show that they do quite well.
- Details about how data is processed for image classification is missing. Do the experiments use embeddings obtained from ResNet on images to get a set of vectors and labels as context, or are the models given raw pixel information in some way?
- While Figure 1 is interesting in showing different models undergo learning in different ways, is there any commonality here? For example, do these trends hold over all kinds of tasks considered?
- A lot of the performance of transformer-based models depends on how the data is represented in the context. Providing additional implementation details about how data is represented for each experiment as well as how training is done would be helpful.
- There is also a lot of interesting ablations in the appendix of the paper on different kinds of transformers (eg. Decoder only vs Encoder-Decoder). Insights about differences between such models is more fundamental than differences between small architectural changes between different named architectures, and thus a detailed comment on them in the main section would be nice.
- The authors should consider citing the following works which are directly related to ICL.

**Citations missing**

Garnelo, M., Schwarz, J., Rosenbaum, D., Viola, F., Rezende, D. J., Eslami, S. M., & Teh, Y. W. (2018). Neural processes. arXiv preprint arXiv:1807.01622.

Garnelo, M., Rosenbaum, D., Maddison, C., Ramalho, T., Saxton, D., Shanahan, M., ... & Eslami, S. A. (2018, July). Conditional neural processes. In International conference on machine learning (pp. 1704-1713). PMLR.

Müller, S., Hollmann, N., Arango, S. P., Grabocka, J., & Hutter, F. (2021). Transformers can do bayesian inference. arXiv preprint arXiv:2112.10510.

Mittal, S., Bracher, N. L., Lajoie, G., Jaini, P., & Brubaker, M. A. (2023, July). Exploring Exchangeable Dataset Amortization for Bayesian Posterior Inference. In ICML 2023 Workshop on Structured Probabilistic Inference {\&} Generative Modeling.

**Questions:**

- *Thus, each train and test example is a unique learning problem, but of a consistent type (e.g. linear regression)* : In this particular setup, “consistent type” does not imply linear regression but just regression in general. Would that be correct?
- The difference between "bursty" and "non-bursty" prompts was not clear, at least to me, from the write-up. Could the authors clarify this distinction?
- For Transformer based experiments (eg. BERT, GPT2, etc.), are positional encodings used? This is primarily because observations seen in linear regression, for eg., are exchangeable and thus observation i and j should not be treated differently (i.e. there should be permutation invariance in the predictive distribution).
- For the transformer models, can the authors clarify what an encoder-only transformer model is?
- It is not clear why the ``ICL Score” is a good metric to care about. Could the authors explain what is the relevance of this score, and why is it important other than validation loss?

---

> ### Author Response · Authors · 2023-11-18
>
> W1:
>
> We agree with the reviewer and have actually reorganized all our experiments, including putting content exclusive to language modeling into Section 6. We did end up removing mentions of micro/macro perspective as we realized it was not strictly needed to understand our draft and seemed to add confusion rather than clarity to other readers.
>
> W2:
>
> We’ve expanded Section 2 to clarify the differences among the various architectures. That said, some architectures, in particular those inspired by state-space models, are quite complex and require substantial background knowledge. In our experimental results, we also spend more time discussing our hypotheses on why we see differences among architectures.
>
> W3:
>
> Perhaps this is a misunderstanding. We also find that decoder-only transformers can do ICL considerably well which is the same design used by Garg et. al (2023). In fact, our revised experiments show GPT2 and Llama2 are often the top performers when given prompt lengths seen during training. The transformers which did underperform in our previous draft were BERT and T5. The prior is an encoder-only transformer and the latter is an encoder-decoder. We are, admittedly, not familiar with Muller et. al (2022). They seem to use a modified encoder-only transformer which is interesting given the poor results we saw with BERT.
>
> In order to improve the focus of our new draft and for practical reasons, we removed experiments for architectures not suitable for causal language modeling (the setting where in-context learning is typically used in the real world). This includes BERT and Fnet. We also removed T5 which does not treat subsequences of its encoder input as training examples. Both types of architectures are not amenable to our experimental setups in Sections 4 and 6. We considered keeping them in Section 5 but decided showing their performance in only a single setting did not add enough value.
>
> In case the reviewer finds this interesting, we did perform additional experiments on encoder-decoder transformers (T5) and encoder-only transformers (BERT) that did not make it into our latest revision. First, we find that how prompts are presented to an encoder-decoder transformer make a big difference. In our original experiments, the prompt is given only to the encoder stack. The decoder stack receives only a single learned start token. This implies that the network has only bidirectional attention across the prompt via its encoder. However, we see significant performance improvements when we pass the prompt to both the encoder and decoder stacks, using only the final prediction of the decoder stack to compute loss. It is still unclear to us why this change matters. Given that we know that decoder-only transformers perform well, perhaps the encoder representations are not useful and are effectively being ignored. An alternative hypothesis is that passing the prompt to the decoder stack simply adds more computation steps which is helpful for ICL to emerge. In regards to the underperformance of encoder-only transformers, we found that this was isolated to the Huggingface implementation of BERT (which we used). We found performance significantly improved when using the x-transformers implementation of an encoder-only transformer. We are still investigating why this is the case.
>
> W4:
>
> We’ve expanded Section 5 to be more explicit about how the image data is being processed. Specifically, we use a randomly initialized Resnet to embed images (raw pixels) into vectors. The image labels are mapped to vectors with a standard embedding layer. The embedded images and labels are then interleaved into a sequence of vectors and passed to the model being evaluated. Resnet and the label embedding layer are trained alongside the model being evaluated.
>
> W5:
>
> We made substantial changes in how we present and discuss the results of our experiments. We provide a brief summary of the changes in the general response but hope the reviewer finds value in our new presentation in Sections 4, 5, and 6.
>
> W6:
>
> We agree with the reviewer and expanded our discussion of experimental details. Specifically, we’ve added more information about how the various tasks are ultimately mapped to a sequence of vectors in Section 2. We included more details surrounding the training and evaluation process in Appendix A.1, A.2 and Section 5.
>
> W7:
>
> Since we made substantial revisions to our draft to improve the focus of our manuscript, we decided to remove most of the appendix experiments found in the original draft. Instead, we spend much more time deeply investigating our new results in Sections 4, 5, and 6.
>
> W8:
>
> Thank you for bringing these works to our attention. We will review these works and figure out how best to incorporate them into our draft. The works appear to represent an interesting perspective on ICL that we had not considered.

---

> > ### Author Response · Authors · 2023-11-18
> >
> > Q1:
> >
> > By “consistent type” we just mean that each prompt (train, test examples) are members of the same class of functions. For example, in our linear regression task, let's say we generate a prompt which is associated with some implicit w_1 (defining a specific instance of a linear regression problem). Now we generate another prompt which is associated with w_2. Both prompts are unique problems (different w) but of the same type: linear regression. That said, you are correct in that the space of predictions by the model is not strictly linear.
> >
> > Q2:
> >
> > We’ve revised Section 5 to explain the differences between “bursty” and “non-bursty” more clearly. We’ve also included visual examples in Table 1 and Figure 6.
> >
> > Q3:
> >
> > Yes, all transformers in our experiments use positional encodings. To be precise, GPT2 uses absolute learned positional embeddings (just another embedding layer) and Llama2 uses rotary positional embeddings. You are correct in that there should be permutation invariance among the input-output pairs. However, the pairs themselves are not permutation invariant as the model needs to be able to distinguish between the input from the output. Still it would be interesting to experiment with positional embeddings removed as prior work has shown that they may not be necessary in causal transformers [1, 2].
> >
> > Q4:
> >
> > An encoder-only transformer is a “BERT-style” transformer, i.e., a single stack of transformer blocks that do not employ causal masking.
> >
> > Q5:
> >
> > ICL score can be thought of as a simple heuristic for measuring statistical efficiency. It differs from validation loss in that it’s effectively the slope of the validation loss curve plotted against token index (context length). Prior work [3] has used ICL score as a signal for when “induction heads” develop, a pair of attention heads that are theorized to be responsible for ICL ability in transformers. We include the metric in our study to see if a similar signal emerges in non-transformers and to gauge whether it correlates well to other measures of ICL ability.
> >
> > [1] https://arxiv.org/abs/2203.16634
> >
> > [2] https://arxiv.org/abs/2305.19466
> >
> > [3] https://arxiv.org/abs/2209.11895

---

> > > ### Comment · Reviewer_9CgZ · 2023-11-21
> > > **Official Comment by Reviewer 9CgZ**
> > >
> > > Thanks to the authors for providing the clarifications to all my questions. While most of my concerns have been addressed, I will choose to keep my current score for the following reasons:
> > >
> > > - **Conflicting results with Muller et. al (2022)**: The code for the mentioned related work is available online, and thus an analysis into why the findings are different would make the authors' case stronger here as well as further understanding into when ICL happens in such architectures and when not.
> > >
> > > - **We also removed T5 which does not treat subsequences of its encoder input as training examples**: I did not understand what this statement meant.
> > >
> > > - **Difference in BERT and T5 performance when using different libraries**: I appreciate the authors' in investigating the source of the incapability of such models demonstrating ICL capabilities but I believe that a more in-depth analysis into the same is useful and needed.
> > >
> > > - **Positional Encoding**: It would also be a good ablation to study the exact same setting with and without positional encodings through different kinds of tokenization schemes. For example, a single token could be the concatenation of the (input, label) pair and then one can forego the use of positional encodings. An analysis along these lines would also help understand if such invariances play a huge role in understanding in-context learning or not.
> > >
> > > I do appreciate the authors' attempt at making the draft much more readable but will keep my current score in light of the above points. I think the paper requires a bit more complete investigations into the different directions laid out.

---

> ### Author Response · Authors · 2023-11-22
>
> We agree that the proposed positional encoding experiment is interesting. We conducted the experiment and included the results in Figure 9 of our latest revision. The setup and our observations are included in the general response. We now address the reviewer’s remaining concerns.
>
> Our findings do not contradict Muller et. al (2022) [1].
> 1. Prior to conducting the aforementioned experiment, our revised draft made no claims about encoder-only transformers (the base architecture used by [1]).
>
> 2. If the reviewer is referring to the results in our original draft (Table 2), there are a few variables at play.
>     - First, the experimental set up of [1] is different from ours.  [1] sums the embeddings of each in-context example pair to represent it as a single token. They also use a special Riemann head for regression tasks and remove positional embeddings.
>     - Second, the poor performance was isolated to the Huggingface implementation of BERT, but we did find that BERT performed better with more hyperparameter tuning (Table 14, original draft). Our current position is that this implementation of BERT might not be a good representative of a basic encoder-only transformer as it was designed to specifically support masked language modeling and next sentence prediction [2]. In our original draft, we found that a generic implementation of an encoder-only transformer performed well at ICL (Table 9, original draft) which does not contradict [1].
>     - Observing that a specific (and popular) implementation of an encoder-only transformer failed at our tasks is valuable in itself. At the time of writing, BERT is the 2nd most downloaded model on the Huggingface hub. It is reasonable to assume that others are using this implementation as a generic encoder-only transformer. We also do not believe that extensively comparing two specific implementations of an encoder-only architecture is within the scope of this study which seeks to compare fundamentally different architecture classes.
>
> 3. Most importantly, in light of the new experiments, our findings are indeed consistent with [1]. Our experimental set up is now comparable and we confirm that encoder-only transformers perform well on all tasks considered.
>
> Questions regarding T5.
> 1. Clarity on “We also removed T5 which does not treat subsequences of its encoder input as training examples”. Apologies for wording this in a confusing way. We will lay out a concrete example. Consider trying to use T5 for the experiments in Section 4. For each architecture, we report performance with respect to context length. We can do this efficiently because the considered architectures are all auto-regressive: each in-context example and its left context is treated as a single input sequence. This is what we meant by “subsequence”. This implies that all in-context examples produce a valid loss and prediction conditioned on its left context. However, this is not straightforward with T5 because its encoder stack has bidirectional attention. Even though we compute loss only using its decoder-stack, cross-attention to the encoder allows the model to “cheat”. This is not to say that Figure 1 is impossible to produce with T5, it is just impractical.
> 2. The initial poor performance of T5 was not due to any specific implementation. We explained earlier that T5 only performed poorly if the prompt is only given to its encoder stack and the decoder stack receives a single learned start token. Performance improves dramatically when the prompt is given to both the encoder and decoder stack. See Table 11 of the original draft where we perform an ablation to confirm this.
> 3. Note that we make no claims about T5 or encoder-decoder transformers in our revised draft. We focus solely on the autoregressive architectures, the dominant paradigm where ICL is used.
>
> We thank the reviewer for engaging in thoughtful discussion and providing constructive feedback. We have made every effort to address their concerns within the rebuttal period and hope that we succeeded.
>
> [1] https://arxiv.org/abs/2112.10510
>
> [2] https://arxiv.org/abs/1810.04805

---

### Official Review · Reviewer_n4qL · 2023-10-31

**Soundness:** 3 good
**Presentation:** 3 good
**Contribution:** 2 fair
**Rating:** 6
**Confidence:** 3

**Summary:**

This paper conducts large-scale experiments to investigate the ability of different network architectures (e.g., RNNs, CNNs, Transformers, and state-space models) to conduct in-context learning. This paper evaluates fifteen architectures on five synthetic in-context learning tasks, and demonstrate that all considered architectures can conduct in-context learning in some cases. In addition, this paper discusses how the causal masking in transformers and the training dynamics of different models influence the in-context learning ability.

**Strengths:**

1.	The paper conducts extensive experiments and provide full details of the training protocols and hyperparameters.
2.	Most part of the paper is easy to read.

**Weaknesses:**

1.	My main concern is that the paper lacks the discussion of under which conditions a certain architecture tends to succeed in in-context learning and under which conditions it fails. In other words, I think the current main claim of this paper that “all considered architectures can perform in-context learning under certain conditions” is not very interesting and surprising. Instead, presenting both the success and failure cases and carefully comparing the settings/conditions for failure cases to occur may provide more insights to the community. I do not expect the paper to find the essential reasons for a certain architecture to fail in in-context learning, but more ablation studies on various training settings/hyperparameters are encouraged and at least some reasonable hypotheses can be made to explain the failure cases. The authors have some discussions on the unexpected low performance of BERT and T5, which I greatly appreciate. However, this is not enough, and I hope to see a more comprehensive picture of how various factors affect the in-context learning ability.
2.	When introducing the image classification task on Page 4, the definition of “bursty” and “non-bursty” samples are unclear and not self-contained. I feel confused without referring to the original paper. I encourage the authors to give a toy example in a figure along with the description.

**Questions:**

1.	What is the difference between the micro perspective of in-context learning and the macro perspective of in-context learning? On Page 4, the paper says “While the micro perspective focuses on specific tasks, the macro perspective offers a broader view of in-context learning by focusing on loss”, but I’m not able to capture the essential difference of the macro perspective from the micro perspective from this sentence.
2.	In the task of Associative Recall, I’m concerned with the following case. If the query token never appears in the prompt sequence, how can the network be able to predict the corresponding target of this query token? For example, if the prompt is “a,1,b,3,c,2,d”, how can the network predict the target for the token “d” when the network has never seen “d” in the prompt?
3.	For CNNs, why do the authors use lightweight convolutions and dynamics convolutions for experiments? Why not consider traditional convolutions, which are more widely used and studied? The authors are encouraged to clarify the reason for this specific setting.
4.	What is the criterion to assign the cells red/green colors in Table 2?

---

> ### Author Response · Authors · 2023-11-18
>
> W1:
>
> We acknowledge that the claims in our original draft were overly conservative. In our revised draft, we demonstrate that all architectures are capable of performing in-context learning across a wider range of conditions than previously documented. Specifically, we highlight how ICL emerges in RNN and LSTM for image classification, which contrasts with the findings of [1].
>
> Furthermore, we have significantly revised our experimental setup and discussion. In Section 4, we report how ICL ability responds to varying prompt lengths, including those not seen during training. In Section 5, we show how changes in the distributional properties of training data affect the emergence of ICL. In Section 6, we focus on tasks involving natural language which more closely resemble ICL in the real world. We also converted our dense data tables into line plots which make relative performance takeaways much easier to see. These revisions allow us to more clearly delineate the circumstances under which each architecture succeeds or fails, as well as to make stronger relative comparisons.
>
> In our revised draft, we have expanded the discussion to include additional hypotheses that may explain the varying performances of different architecture. While we recognize the importance of understanding the 'why' behind these performance differences, we believe that first clearly documenting the observed phenomena is an equally important contribution to the research community.
>
> W2:
>
> We’ve revised Section 5 to explain the differences between “bursty” and “non-bursty” more clearly. We’ve also included visual examples in Table 1 and Figure 6.
>
> Q1:
>
> We ended up removing mentions of the micro and macro perspective in our revised draft. These perspectives were introduced by [2] but we now believe that, in the context of this study, end up adding confusion rather than clarity.
>
> Q2:
>
> You are correct. If the query token never appears in the prompt sequence, the network cannot accurately predict the corresponding target. However, all architectures are trained and evaluated with the same datasets so their performance ceiling is the same, i.e., we can still make relative comparisons among architectures. We also find that given a modest prompt length, the probability of the query token not appearing in the prompt decays quickly. We can see in Table 9 that prompts of length 2^5 to 2^6 have scores near 100% accuracy, indicating that the query must have almost always appeared in those prompts.
>
> There are also practical concerns with forcing the query and its completion to appear in the prompt as it makes performing the experiments where we show how accuracy changes with varying prompt length difficult (Figure 1). This introduces a dependence on what may appear in the in-context examples and the current query index we are computing loss for.
>
> Moreover, we’ve found that this task is present in prior work on ICL [3] [4] so we felt it important to include in our study for completeness.
>
>
> Q3:
>
> We originally considered adding traditional convolutions to our experiments but decided to only focus on architectures capable of causal language modeling (the setting where in-context learning is typically used in the real world). There are also practical concerns that arise when conducting our experiments. Specifically, the experiments in Section 4 where we examine performance versus varying prompt lengths as well as the language modeling experiments in Section 6 are not straightforward to perform with a traditional CNN.
>
> Given our criteria, we are not aware of a CNN based architecture that is simpler than lightweight convolutions. It is simply a (normalized) depthwise convolution which has been widely studied [5]. There exists Gated Convolutional Model [6] which predates lightweight convolutions but is arguably more “complex”. That said, if the reviewer has a specific architecture mind, we are happy to perform and add the experiments to our final draft. However, given that we’ve increased the number of training runs substantially for each architecture, the experiments may not finish before the rebuttal period.
>
> Q4:
>
> We have since removed the red/green coloring in the data tables and replaced them with line plots for clearer presentation of the data. A new view of the dense data tables are still available in the appendix. The color coding in these new data tables are simply normalized with respect to the min and max values within that table.
>
> [1] https://arxiv.org/abs/2205.05055
>
> [2] https://arxiv.org/abs/2209.11895
>
> [3] https://arxiv.org/abs/2212.14052
>
> [4] https://arxiv.org/abs/2302.10866
>
> [5] https://arxiv.org/abs/1610.02357
>
> [6] https://arxiv.org/abs/1612.08083

---

> > ### Comment · Reviewer_n4qL · 2023-11-21
> >
> > I want to thank the authors for their response and revision of the paper. The presentation of experimental results in the revised paper becomes much clearer than the previous version, and I see some new discussions about failure cases and the "extrapolation" regime. I have raised my score accordingly.
> >
> > One concern is that the revised paper seems significantly different from the original paper (although I appreciate many of the changes made to the paper and the newly conducted experiments). I would appreciate it if the authors could ensure that the results of the new experiments (which may be conducted in a hurry) are sound and reproducible.

---

> > > ### Author Response · Authors · 2023-11-23
> > >
> > > Thank you for taking the time to consider our revisions. We have made every effort to ensure that the results are sound and reproducible.

---

### Official Review · Reviewer_Rxok · 2023-11-01

**Soundness:** 2 fair
**Presentation:** 3 good
**Contribution:** 1 poor
**Rating:** 5
**Confidence:** 2

**Summary:**

The paper is an empirical study investigating the ability to perform in-context learning of different architectures. It carries out the study with tasks in the form of few-shot learning: a prompt with a few input-output pairs and a final input query; the model outputs the prediction of the query. There are 5 tasks proposed: associative recall, linear regression, multiclass classification, image classification and language modeling, where the first 3 tasks use synthetic data. A wide range of model architectures are studies, including RNN, Convolutional networks, transformers and state space models. The conclusion is that all architectures exhibit in-context learning capabilities when tuned appropriately.

**Strengths:**

The paper presents a study that includes a wide range of architectures and tasks. The implementation looks reproducible and results seem to be credible to me.

**Weaknesses:**

Is the conclusion of the paper trivial? The few-shot learning task still can be seen as having a mapping between input and output, albeit the input are more complicated in this case. So I would expect that any universal approximator should at least have the capability to perform such a task. Of course, the performance would vary because of characteristics of the architecture and tuning. But the authors don't seem to put emphasis on the relative performance of the different architectures in their study.

**Questions:**

1. It sounds like in-context capability comes from the the fact that data are presented in a sequence and models are able to make use the temporal correlation between tokens when doing predictions. Has the authors investigated what order of the sequence encourages the learning of the correlations?

2. Why do you choose to only reduce depth when normalize the parameter counts? I thought normal practice is to reduce both the width and depth in some propotion.

---

> ### Author Response · Authors · 2023-11-18
>
> While we agree that any theoretical universal approximator should have the capability to learn any input-output mapping, this may not be true in practice. Indeed, prior work has shown that it is not clear if non-transformer architectures can perform ICL [1]. That said, we agree with the reviewer that we did not place enough emphasis on the relative performance of the architectures. We have since addressed this in our revised draft by conducting both a larger hyperparameter sweep and revising our experimental set up. Specifically, in Section 4, we report how ICL ability responds  to varying prompt lengths, including those not seen during training. In Section 5, we show how changes in the distributional properties of training data affect the emergence of ICL. In Section 6, we focus on tasks involving natural language which more closely resemble ICL in the real world.
>
> Q1: We agree that investigating the effects of sequence order is interesting as most ICL tasks are permutation invariant with respect to in-context examples. While we do not experiment with sequence order specifically, we do perform a new study on sequence distributional properties in Section 5. That is, we investigate how different sampling procedures used to construct training sequences affect ICL.
>
> Q2. Thank you for pointing this out. Indeed we found that increasing depth for smaller parameter architectures (on a per layer basis) sometimes led to unstable training. We have since redone our experiments and normalized parameter accounts by adjusting both depth and embedding size. We’ve included the specific widths and depths we selected in Tables 3 and 5.
>
> [1] https://arxiv.org/abs/2205.05055

---

> > ### Author Response · Authors · 2023-11-22
> >
> > We hope that the reviewer has found our latest revision much improved. We specifically set out to address their concern that we did not place enough emphasis on the relative performance of different architectures. The reviewer will be happy to find that we thoroughly address this in Sections 4-6.
> >
> > We also believe that the reviewer will find the new experiment suggested by reviewer 9CgZ relevant to their question regarding sequence order. We have outlined the experiment and its results in the general response.

---

> > > ### Comment · Reviewer_Rxok · 2023-11-23
> > > **Thanks for the revision**
> > >
> > > I would like to thank the authors for the revision. The updates significantly improve the paper, both in terms of content and presentation. In particular, I find extending the test sequence longer than the training sequences is more revealing for in-context learning ability. The conclusion of the paper is also more meanfully emphasising the relative performance between different architectures.
> > >
> > > I do share the same concern as the other reviewer that, considering the significant changes made to the paper, it may not be scrutinized enough given the limited time for conducting experiments and for peer review.
> > >
> > > Considering all aspects, I would like to raise by score to 6 (marginally above the acceptance threshold).

---

### Official Review · Reviewer_7V5r · 2023-11-04

**Soundness:** 3 good
**Presentation:** 2 fair
**Contribution:** 3 good
**Rating:** 6
**Confidence:** 4

**Summary:**

The authors probe several different model architectures for their ability to perform in-context learning, a capability primarily associated with the transformer architecture. Prior works have slightly conflicting results on this front: e.g., some works have shown LSTMs/RNNs can perform ICL, while others find that to be incorrect. By normalizing the setup and performing an extensive analysis on a bunch of both synthetic and natural tasks, the authors provide a useful study to address the question of which architectures can perform ICL and also better contextualize the debates in prior works.

**Strengths:**

The paper is primarily of an empirical nature and I found its analysis to be very extensive and impressive. The tasks are valuable, if not novel (to be clear, that is fine), since they cover a broad spectrum of prior works' models for understanding in-context learning in neural networks.

**Weaknesses:**

My primary apprehension revolves around the paper's presentation: The results, at times, read like the authors ran out of time and hence were not able to provide a sufficient discussion of what they observed in the experiments (especially for the results that show up later in the paper, which focus on more realistic tasks). Relatedly, due to the presence of a huge set of experiments in any given table, drawing any conclusions gets quite difficult and the reader has to rely on the authors' writeup. Bar plots would be a better idea than a table for presenting such dense results, since it allows easier comparisons between results.

**Questions:**

As the authors mention in the paper, often efficient transformer variants are used by practitioners, but results / analyses on ICL focus on its dense, quadratic complexity version. Are there any results on, e.g., linear attention's or sparse attention's ICL performance? I think adding these results will be very helpful if possible.

---

> ### Author Response · Authors · 2023-11-18
>
> We agree with the reviewer that our data presentation can be improved. To that end, we converted all the dense tables into line plots which allow us to quickly visualize how ICL performance responds to prompt length and how it changes over the course of training. The dense tables are still accessible in the appendix and we’ve added bidirectional links for easy navigation. We found that patterns in the new visualizations are now much easier to identify. We also reorganized our results into three sections that focus on different aspects of ICL performance: extrapolation, data distributional properties, and natural language. We hope the reviewer finds more value in our revised data visualizations and discussion compared to our original draft.
>
> We are happy to add additional experiments for linear attention and sparse attention in our final draft. Given that we’ve increased the number of training runs substantially for each architecture, the experiments may not finish before the rebuttal period. If helpful, one of the architectures we do have experiments for, RWKV, does employ a linear attention mechanism and can be viewed as a successor to An Attention Free Transformer [1].
>
> [1] https://arxiv.org/abs/2105.14103

---

> > ### Comment · Reviewer_7V5r · 2023-11-22
> >
> > Thank you for the response. The updated presentation of results is indeed much better and makes the paper more legible. I'll keep my score as is.

---

> > > ### Author Response · Authors · 2023-11-23
> > >
> > > Thank you for taking the time to consider our revisions. We understand that your score will remain unchanged. That said,
> > > are there any specific concerns that prevent you from increasing your score? Your feedback is valuable, not only for improving this specific work, but also for guiding our future work.

---

### Author Response · Authors · 2023-11-18
**Response to All Reviewers**

We thank the reviewers for their time and thoughtful feedback. We made substantial revisions to improve the clarity and focus of our draft as well as address a number of your concerns. In particular:

1. Revised experimental settings allow for better relative comparisons among architectures. In Section 4, we report how ICL ability responds  to varying prompt lengths, including those not seen during training. In Section 5, we show how changes in the distributional properties of training data affect the emergence of ICL. In Section 6, we focus on tasks involving natural language which more closely resemble ICL in the real world.

2. Larger hyperparameter sweeps allow for stronger relative claims and flip some failure cases. Specifically, in Sections 4 and 5,  we increased the number of training runs per architecture/task combination from 6 (original draft) to a range of 81 to 144, depending on the specific architecture and task. We’ve also included a more detailed treatment of our experimental set up and training procedure.

3. Improved data presentation. The dense tables, which were difficult to interpret, were converted to line plots for clearer and quicker comprehension of the results. The tables are still available in the appendix and we added bidirectional links for easy navigation to and from the main document and the appendix.

4. Revised results: All architectures demonstrate ICL ability in most conditions, even those where prior work found they failed. However, in Section 4,  we find stark differences in their statistical efficiency and ability to extrapolate to long prompts lengths not seen during training. Section 5 highlights architectures that demonstrate a strong inductive bias towards ICL, even when presented with alternative learning routes. Finally, in Section 6, we discover that evidence used in prior work to support a mechanistic interpretation of how ICL forms in transformers is also present across most architectures that lack an attention mechanism.

---

> ### Author Response · Authors · 2023-11-22
>
> As the rebuttal period comes to a close, we would like to thank the reviewers again for their time and constructive feedback. Reviewer 9CgZ proposed an interesting experiment to measure the effects of positional embeddings given that in-context examples in our tasks should be permutation invariant. This experiment may also be of interest to reviewer Rxok who asked about the effects of sequence order. We conducted this experiment and show the results in Figure 9 of the latest revision (page 29).
>
> Specifically, we consider the following variables:
> - Token representation scheme: We represent in-context example pairs as a single token (instead of two in our original experiments) which allows us to remove positional embeddings. Specifically, we either sum their respective embeddings as represented in [1] or concatenate their embeddings, as suggested by reviewer 9CgZ. The query label is masked out by setting its embedding to zero.
> - Positional embeddings: whether to use learned absolute positional embeddings or no positional embeddings at all.
> - Attention mask: encoder-only vs decoder-only transformer. Note that in both scenarios, the query can attend to all in-context examples. In the encoder-only transformer, each example can attend to all other examples since it does not employ a causal mask. Examples in the decoder-only transformer can only attend to examples to its left.
> - 4 tasks: linear regression, associative recall, multiclass classification, image classification.
> - The remaining settings are identical to Section 4 with the following changes: Our hyperparameter sweep covers 2 learning rates, 2 seeds, and 2 layer depths. We train for 50K steps and only take the loss (and evaluate) at the token index 32 (i.e., models are trained to make a single prediction given 31 example pairs and the query). We conducted 768 training runs in total.
>
> We make the following observations:
> - Token representation scheme sensitivity: Associative recall and multiclass classification are not sensitive to tokenization schemes. However, we observe that concatenating embeddings in linear regression and image classification resulted in noticeably improved performance. We suspect that it is easier for attention heads to discern in-context inputs from outputs if they initially reside in their own subspace.
> - Removing positional embeddings did not impact performance. This makes intuitive sense as in-context examples in this setting are permutation invariant.
> - For most tasks, encoder-only and decoder-only transformers perform on par. The exception was linear regression where the encoder-only outperformed the decoder-only in the more difficult settings (d=20, 30). For image classification, we observed that ICL emerged in both transformers in very similar windows and followed a similar decay scheduled (as discussed in Section 6).
>
> [1] https://arxiv.org/abs/2112.10510

---

### Meta-Review · Area_Chair_8xKA · 2023-12-10

**Metareview:**

The paper explores the in-context learning (ICL) capabilities of various model architectures, including rnns, conv nets, transformers, and ssms. The study conducts extensive experiments on different architectures across a suite of synthetic ICL tasks. The findings suggest that all considered architectures can perform ICL under certain conditions, with contemporary architectures, particularly ssms, showing robustness in more complex tasks.

Strengths:
- The paper conducts a comprehensive examination of various architectures, covering a wide range of paradigms and tasks.
- It provides new insights into the capabilities of ssms compared to transformers and other architectures.
- Authors revised the presentation format from dense tables to line plots, enhancing clarity and comprehension.
- The authors made substantial revisions and conducted additional experiments to address concerns raised by reviewers.

Weaknesses:
- Initial drafts had issues with the clarity of results presentation, though this was improved in revisions.
- Some reviewers noted a lack of detailed discussion on why certain models perform differently and the impact of positional encodings.
- A more thorough comparative analysis, especially between different transformer implementations and their impact on ICL, could provide deeper insights.

AC votes for the acceptance of the paper.

**Justification For Why Not Higher Score:**

See weaknesses.

**Justification For Why Not Lower Score:**

The papers' findings are novel and interesting to the sequence modeling community.

---

### Decision · Program_Chairs · 2024-01-16

Accept (poster)